# Nearing or Surpassing: Overall Evaluation of Human-Machine Dynamic Vision Ability

## Abstract

Dynamic visual ability (DVA), a fundamental function of the human visual system, has been successfully modeled by many computer vision tasks in recent decades. However, the prosperity developments mainly concentrate on using deep neural networks (DNN) to simulate the human DVA system, but evaluation frameworks still simply compare performance between machines, making it tough to determine *how far the gap is between humans and machines in dynamic vision tasks.* In fact, neglecting this issue not only makes it hard to determine the correctness of current research routes, but also cannot truly measure the DVA intelligence of machines. To answer the question, this work designs a comprehensive evaluation framework based on the 3E paradigm – we carefully pick 87 videos from various dimensions to construct the environment, confirming it can cover both perceptual and cognitive components of DVA; select 20 representative machines and 15 human subjects to form the task executors, ensuring that different model structures can help us observe the effectiveness of research development; and propose multiple evaluation indicators to quantify their DVA. Based on detailed experimental analyses, we first determine that the current algorithm research route has effectively shortened the gap. Besides, we further summarize the weaknesses of different executors, and design a human-machine cooperation mechanism with superhuman performance. In summary, the contributions include: (1) Quantifying the DVA of humans and machines, (2) proposing a new view to evaluate DVA intelligence based on the human-machine comparison, and (3) providing a possibility of human-machine cooperation. The 87 sequences with frame-level human-machine comparison and cooperation results, the toolbox for recording real-time human performance, codes for sustaining various evaluation metrics, and evaluation reports for 20 representative models will be open-sourced to help researchers develop intelligent research on dynamic vision tasks.

## 1 Introduction

Research on visual abilities can be dated back to the last century (Hubel & Wiesel (1959; 1962)). Neuroscientists divide the human visual system into two categories, namely the static vision ability (SVA) to perceive the details of static objects (Chan & Courtney (1996)), and the dynamic vision ability (DVA) to track moving objects (JW et al. (1962)). These two visual abilities are essential in our daily life (Land & McLeod (2000); Beals et al. (1971); Burg (1966); Kohl et al. (1991)), and have been modeled by a series of computer vision tasks. Recently, with the growth of dataset scale and the abundance of computing resources, most data-driven algorithms achieve higher and higher scores in experimental environments, and are widely employed in various scenarios (Dankert et al. (2009); Weissbrod et al. (2013); Wei & Kording (2018)).

However, some bad cases hidden under the prosperity development challenge the state-of-the-art (SOTA) algorithms. For example, visual models usually decrease their perception when encountering unknown-category targets under special illumination conditions (*e.g.*, a vision-based autonomous vehicle crashes into a large truck at night). This shortcoming is far from humans' powerful visual abilities and may cause safety hazards, causing us to rethink – with the support of massive datasets and powerful computing resources, why can't SOTA models achieve a similar visual ability to humans? Do existing evaluation methods actually measure the visual intelligence of machines?

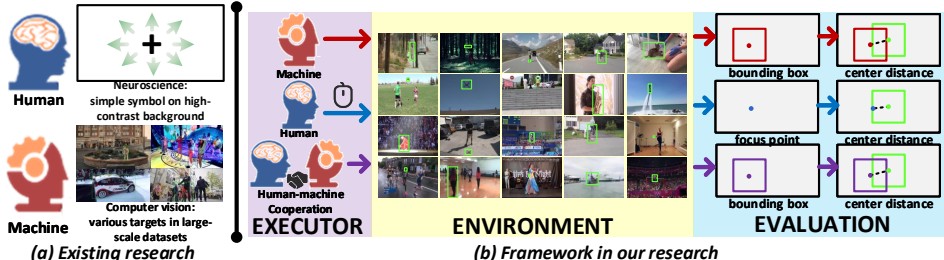

Figure 1: Comparison of the evaluation frameworks in dynamic vision abilities. (a) Existing research on DVA evaluation for human and machine is entirely separate. (b) We follow 3E paradigm (Hu et al. (2022a)) to design an overall evaluation framework of human-machine DVA.

Compared to thriving algorithm research, evaluation methods have received far less attention. Traditional evaluations mainly focus on the performance comparison between machines (Deng et al. (2009); Russakovsky et al. (2015)), causing the goal to become *downward* (*i.e.*, a method that exceeds all others is considered excellent). In fact, these evaluation mechanisms actually measure the machine's performance rather than intelligence. When we refer to machine intelligence, a natural association is Turing test (Turing (2009)), which indicates that machine intelligence evaluation requires human participation, and has gradually attracted scholars to propose a series of essential works in decision-making problems (*e.g.*, AlphaGo (Silver et al. (2017)) and DeepStack (Brown & Sandholm (2018))). Using humans as a reference, the goal will become *upward* – exemplary machines must gradually move closer to human capabilities. In other words, *exploring the gap between humans and machines is very crucial for machine intelligence evaluation*.

Based on this idea, recent works have introduced humans to several vision tasks, using human baseline to measure machine intelligence. Some researchers endeavor to analyze the gap between machines and humans in image classification (Geirhos et al. (2018; 2020; 2021)), others investigate the attention areas in static images to explore visual selectivity (Langlois et al. (2021)). Regrettably, their research scope is mainly restricted to static vision tasks, while neuroscience and cognitive psychology studies have shown that the correlation between SVA and DVA is naturally low (Long & Penn (1987)). On the other hand, existing research on DVA evaluation for human and machine is entirely separate. As shown in Figure 1 (a), neuroscientists use toy environments to measure human DVA (Pylyshyn & Storm (1988)), while computer scientists evaluate machines through large-scale datasets (Fan et al. (2020); Huang et al. (2021)) – neither the environment nor the evaluation mechanism are compatible, causing the comparison of human-machine DVA to be impossible. Therefore, an intuitive question is, *how to compare humans and machines in dynamic vision ability?* To answer this question, we should first select a suitable computer vision task to represent human DVA, then design a evaluation framework to accomplish human-machine DVA measurement and comparison.

Compared with other tasks like multi-object tracking (MOT) (Ciaparrone et al. (2019)) and video object detection (VOD) (Wang et al. (2018)), single object tracking (SOT, *i.e.*, locates a user-specified moving target in a video) (Wu et al. (2015)) is a *category-independent* task with no constraint on motion continuity, scene change, or object category, and can be regarded as the closest task to represent human DVA (Appendix A.1). Due to the human-like task definition of SOT, excellent task executors should not only keep tracking the moving target (*perception*) but also re-locate the target when its position is mutated (*cognition*) (Appendix A.2). From above analyses, we select SOT as the representative task, and follow 3E paradigm (Hu et al. (2022a)) to design an overall evaluation framework (Figure 1). The technical difficulties and our contributions are as follows.

**Experimental environment construction (Section 2.1).** The first difficulty in environment construction is *compatibility*. Choosing a high-contrast toy environment used in classical neuroscience work is too simple to evaluate DVA accurately. On the other hand, human psychophysical experiments are expensive and time-consuming, and cannot be assessed on large-scale datasets like machines. Based on this, the second difficulty is *representativity*. With the limitation of dataset scale, the environment should not only fully represent the characteristics and difficulties of DVA, but also provide a graded experimental setup for subsequent analyses. As shown in Table 1, to entirely reflect the task characteristics and thoroughly compare the human-machine DVA, we choose 87 videos

with 244,455 frames to construct the environment. All videos are carefully picked from various dimensions, ensuring the environment can cover the perceptual and cognitive components of DVA.

**Experimental executor selection and record (Section 2.2).** As an interdisciplinary experiment, the task executors involve humans and machines. The difficulty for human subjects is *accurately recording their performance* in dynamic vision tasks and ensuring that the recorded results can be quantified to support subsequent evaluations. Through detailed comparison and analysis, we select the mouse-based method and design a toolbox for human subjects to record their frame-level performance in real-time. For machines, since we aim to study the DVA gap between humans and machines, there are two concerns in model selection: *trends* (*i.e.*, selecting various algorithms) and *gap* (*i.e.*, focusing on the upper bound of algorithms). Thus, both classic and SOTA methods with different architectures are selected to explore whether the research route has effectively shortened the gap. Consequently, task executors included 20 representative algorithms and 15 subjects (all subjects are computer vision researchers aged 20-30). All experiments are managed in a strict process .

**Evaluation metrics design (Section 2.3).** Traditional indicators usually select the positional relationship (*e.g.*, intersection over union (IoU) and center distance) between predicted result $p_t$ and ground-truth $g_t$ to accomplish calculation. However, the tracking result of humans is a point, which cannot be calculated by IoU (Zheng et al. (2020); Rezatofighi et al. (2019)). Thus, we use two center points (*i.e.*, the target center $c_p$ predicted by executors, and the center point $c_g$ of ground-truth) to design three granularities (frame-level, sequence-level, and group-level) for evaluation. Especially, we consider the influence of sequence length and revise the group-level indicators to generate a more appropriate evaluation. Experimental analyses have verified the validity of proposed metrics.

Based on above steps, we quantify the DVA of executors and conduct detailed comparisons (Section 3). By analyzing and summarizing the results, we find that: (1) **Synthetically, humans have stronger dynamic vision ability than machines.** Experimental results demonstrate that humans outperform machines when both perceptual and cognitive abilities are required. Particularly, when we mainly examine the perceptual ability in dynamic vision tasks, we also notice that **the gap of human-machine perceptual ability is closing.** (2) **Human-machine cooperation is possible and effective in dynamic vision tasks.** Based on human-machine differences in perception and cognition, we design a set of cooperative experiments. Results indicate that the SOTA machines with human-machine cooperation can achieve even better performance than humans in some cases.

In summary, this work starts with designing a more intelligent evaluation framework, measures the DVA gap between humans and machines, determines the effectiveness of current algorithm research routes, finds the weaknesses of different executors, and notes the possibility of human-machine cooperation. We will open-source the toolbox, code, and metrics used in the experiments to help researchers further develop intelligent research on dynamic vision tasks.

## 2 METHODS

### 2.1 ENVIRONMENT: DATASETS

For DVA evaluation, researchers initially require subjects to observe moving objects on a high-contrast background. For example, an early work displays some moving plus (+) signs on the screen(Pylyshyn & Storm (1988)). Recently, researchers evaluate the DVA of athletes based on a software named DynVA (Quevedo et al. (2012)), which includes various stimuli with different color and motion trajectories. Although above works have developed from simple designs to computer programs, they all have limited target categories (*e.g.*, specific symbols), simple backgrounds (*e.g.*, colors or some static photographs), and limited motion patterns (*e.g.*, lateral, vertical, and oblique).

Unlike above toy environments that track simple symbols in a high-contrast background, our environment fully models the application scenes and designs various data combinations for ability analyses (*e.g.*, perception, cognition, and robustness). Concretely, the environment should consider the diversity of video contents and includes the variation of video length (especially long sequences). Thus, we refer SOTVerse (Hu et al. (2022a)) and select sequences from representative benchmarks (OTB (Wu et al. (2013; 2015)), VOT series (Kristan et al. (2013; 2014; 2015; 2016; 2017; 2018; 2019)), GOT-10k (Huang et al. (2021)), LaSOT (Fan et al. (2020)), and VideoCube (Hu et al. (2022b))) to form the experimental environment (Table 1 and Appendix C.1).

Table 1: Information on environment settings.

| Task Settings | | | Characteristics | | Ability | Group | Frames |
|---|---|---|---|---|---|---|---|
| | | | Target Absent | Shot-cut | | | |
| **Short-term tracking** (Target presents from beginning to end) | | | N | N | Perception | A | 500-1,000 |
| | | | | | | B | 1,000-2,000 |
| **Long-term tracking** (Target may disappear and reappear in a single shot) | | | Y | N | Perception and cognition | C | 1,000-2,000 |
| | | | | | | D | 5,000-10,000 |
| **Global instance tracking** (Target may disappear and reappear in multiple shots) | | | Y | Y | | E | 1,000-2,000 |
| | | | | | | F | 5,000-10,000 |
| | | | | | | G | 15,000-30,000 |
| **Short-term tracking with challenging factors** (Target presents from beginning to end) | Challenging factors in single frame | Abnormal ratio | N | N | Perception and robustness | H | 500-1,000 |
| | | Abnormal scale | | | | I | |
| | | Abnormal illumination | | | | J | |
| | | Blur bounding-box | | | | K | |
| | Challenging factors between consecutive frames | Drastic ratio variation | | | | L | |
| | | Drastic scale variation | | | | M | |
| | | Drastic illumination variation | | | | N | |
| | | Drastic clarity variation | | | | O | |
| | | Fast motion | | | | P | |
| | | Low correlation cofficient | | | | Q | |

## 2.2 EXECUTOR: MODELS, HUMAN SUBJECTS

As shown in Table 2, 20 represent machines covering both classic and SOTA methods are selected: 2 CF-based trackers (Henriques et al. (2015); Danelljan et al. (2017)), 10 SNN-based methods (Bertinetto et al. (2016); Li et al. (2018b); Zhu et al. (2018); Li et al. (2018a); Yan et al. (2019); Zhang & Peng (2019); Guo et al. (2020); Xu et al. (2020); Zhang & Peng (2020); Voigtlaender et al. (2020)), 5 algorithms that combine CF and SNN (Danelljan et al. (2018); Bhat et al. (2019); Danelljan et al. (2020); Mayer et al. (2021)), and 3 custom networks (Huang et al. (2019); Bhat et al. (2020); Cui et al. (2022)). Please refer to Appendix B for technical details.

For humans, we start the measurement of human DVA based on the following concerns:

**Organization.** 87 videos with different durations, object classes, and scene categories are played at 25FPS to 15 subjects (aged between 20-30, all computer vision researchers) under a high-quality experimental organization – this organization is known as *small-N design* (Smith & Little (2018)), and has been widely used in SVA experiments (Geirhos et al. (2018; 2020; 2021)).

**Approval.** We have obtained the approvals of all subjects. They all have signed an experimental statement (the template is provided by our IRB, which includes experiment purpose, procedure, risks and discomforts, costs, and confidentiality (Appendix C.2)) before the experiment.

**Process.** Previous works mainly use eye-tracker Hu et al. (2022b); Xia et al. (2021) or mouse Geirhos et al. (2018; 2020; 2021) to record human visual ability. To select a better measurement method, all subjects tried both mouse and eye-tracker, and they express that the mouse is better. Besides, we have compared these two devices from practical and theoretical views in Appendix C.3, and finally select the mouse-based method. Experiment includes four steps (Appendix C.4): (1) Subjects check instruments; then adjust the seat height, sitting posture, and the distance to the screen. (2) 2 TEST videos occur sequentially in the screen center. Subjects observe and remember the target features in the first frame, then move the mouse to track the target in subsequent frames. (3) 17 FORMAL videos belonging to different groups are played sequentially (a rest time occurs between two videos to relieve visual fatigue); subjects should concentrate on the target and maintain the mouse's position. (4) Subjects fill in a questionnaire to self-evaluate their performance (Appendix C.5).

## 2.3 EVALUATION: METRICS

This work evaluates executors under the one-pass evaluation (OPE) mechanism, which initializes an executor in the first frame and continuously records the tracking results. For the $t$-th frame $F_t$ in a sequence $s_i = \{F_1, F_2, \ldots, F_t, \ldots\}$, we support the predicted result is $p_t$, the ground-truth is $g_t$, and their center points $c_p$ and $c_g$ are used to design indicators. Note that target absent is regarded as an empty set (*i.e.*, $g_t = \phi$) and excluded by the evaluation process. To accurately evaluate executors' performance, we divide the evaluation dimensions into three granularities, as shown in Figure 2.

Table 2: The performance (based on $NP_{L3}^w$) about human subjects and 20 representative models (SNN-Siamese Neural Network. CF-Correlation Filter. CNN-Convolutional Neural Network. Red, magenta and cyan represent the top-3 machines).

| Executor | Aritciture | Characteristic | Score |
|---|---|---|---|
| Subject_Top | - | The best performance of subjects | 0.891 |
| Subject_Mean | - | The mean performance of subjects | 0.853 |
| Subject_Bottom | - | The worst performance of subjects | 0.801 |
| MixFormer (Cui et al. (2022)) | Custom networks | Transformer-based framework | 0.766 |
| KYS (Bhat et al. (2020)) | Custom networks | Scene information | 0.528 |
| GlobalTrack (Huang et al. (2019)) | Custom networks | Zero cumulative error | 0.645 |
| KeepTrack (Mayer et al. (2021)) | SNN+CF | Target candidate association | 0.718 |
| SuperDiMP (Danelljan et al. (2020)) | SNN+CF | Probabilistic regression | 0.701 |
| PrDiMP (Danelljan et al. (2020)) | SNN+CF | Probabilistic regression | 0.683 |
| DiMP (Bhat et al. (2019)) | SNN+CF | Better discriminative ability | 0.597 |
| ATOM (Danelljan et al. (2018)) | SNN+CF | Combine SNN with CF | 0.506 |
| SiamRCNN (Voigtlaender et al. (2020)) | SNN | Re-detection mechanism | 0.748 |
| Ocean (Zhang & Peng (2020)) | SNN | Anchor-free | 0.635 |
| SiamFC++ (Xu et al. (2020)) | SNN | Anchor-free | 0.512 |
| SiamCAR (Guo et al. (2020)) | SNN | Anchor-free | 0.480 |
| SiamDW (Zhang & Peng (2019)) | SNN | Deeper and wider backbone | 0.558 |
| SPLT (Yan et al. (2019)) | SNN | Local search and global search | 0.610 |
| SiamRPN++ (Li et al. (2018a)) | SNN | Deeper backbone | 0.662 |
| DaSiamRPN (Zhu et al. (2018)) | SNN | Data augmentation | 0.528 |
| SiamRPN (Li et al. (2018b)) | SNN | Region proposal network | 0.495 |
| SiamFC (Bertinetto et al. (2016)) | SNN | Originator of SNN-based trackers | 0.285 |
| ECO (Danelljan et al. (2017)) | CNN+CF | Combine CNN with CF | 0.377 |
| KCF (Henriques et al. (2015)) | CF | Representative CF-based method | 0.270 |

**L1: Frame-level.** *Precision* score (PRE) (Wu et al. (2015)) in frame-level $P_{L1}$ equals the center distance $d_c$. Recently, *normalized precision score* (N-PRE) (Hu et al. (2022b)) is proposed to exclude the influence of target size and frame resolution. Trackers with a predicted center outside the ground-truth will add a penalty item $d_c{}^p$ (*i.e.*, the shortest distance between center point $c_p$ and the ground-truth edge). For trackers whose center point falls into the ground-truth, the center distance $d_c{}'$ equals the original precision $d_c$ (*i.e.*, $d_c{}^p = 0$). Besides, the maximum value in frame $F_t$ is used to normalize the result and generates the final N-PRE score $NP_{L1}$, as shown in Equation 1.

$$P_{L1} = d_c = \|c_p - c_g\|_2$$
$$NP_{L1} = \mathcal{N}(d_c{}') = \frac{d_c{}' - \min(\{d_i{}'|i \in F_t\})}{\max(\{d_i{}'|i \in F_t\}) - \min(\{d_i{}'|i \in F_t\})}, d_c{}' = d_c + d_c{}^p \tag{1}$$

**L2: Sequence-level.** Precision score in sequence-level $P_{L2}(\theta_d)$ is defined as the proportion of frames whose center distance $d_c \leq \theta_d$. To illustrate the performance under different thresholds, previous works (Wu et al. (2015); Fan et al. (2020)) usually draw the statistical results based on different $\theta_d$ into a curve named *precision plot*. Since $\theta_d = 20$ is wildly used to rank trackers, we define $P_{L2} = P_{L2}(20)$ in following experiments. Similarly, draw statistical results based on different $\theta_d{}'$ into a curve generates the *normalized precision plot*. However, we note that directly select a $\theta_d{}'$ to rank executors may introduce human factors, thus we use the proportion of frames whose predicted center $c_p$ successfully fall in the ground-truth rectangle $g_t$ (Flag=Y in Figure 2 (L2)) as $NP_{L2}$. The calculation is listed in Equation 2, where $|\cdot|$ is the cardinality.

$$P_{L2}(\theta_d) = \frac{1}{|s_i|} |\{F_t : d_c \leq \theta_d\}|, P_{L2} = \frac{1}{|s_i|} |\{F_t : d_c \leq 20\}|$$
$$NP_{L2}(\theta_d{}') = \frac{1}{|s_i|} \left|\left\{F_t : \mathcal{N}(d_c{}') \leq \theta_d{}' \in [0,1]\right\}\right|, NP_{L2} = \frac{1}{|s_i|} |\{F_t : c_p \in g_t\}| \tag{2}$$

**L3: Group-level.** For a group of sequences $\mathcal{G} = \{s_1, s_2, \ldots, s_t, \ldots\}$, existing works usually use the mean value of all sequence-level results as the group-level evaluation, as illustrated in Equation 3.

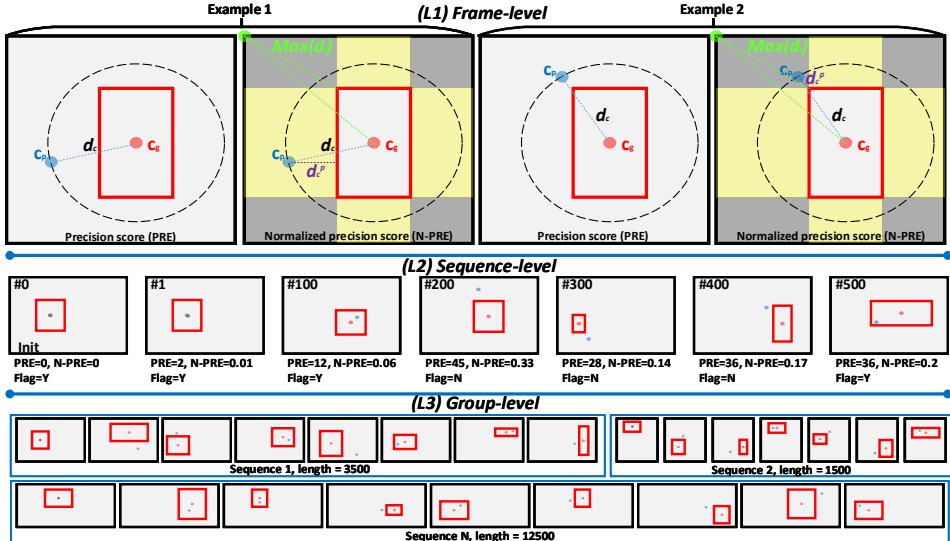

Figure 2: Illustration of three granularities in evaluiation process.

$$P_{L3}(\theta_d) = \frac{1}{|\mathcal{G}|} \sum_{s_i \in \mathcal{G}} \frac{1}{|s_i|} |\{F_t : d_c \leq \theta_d\}|, P_{L3} = \frac{1}{|\mathcal{G}|} \sum_{s_i \in \mathcal{G}} \frac{1}{|s_i|} |\{F_t : d_c \leq 20\}|$$

$$NP_{L3}(\theta_d) = \frac{1}{|\mathcal{G}|} \sum_{s_i \in \mathcal{G}} \frac{1}{|s_i|} \left|\left\{F_t : \mathcal{N}(d_c') \leq \theta_d' \in [0,1]\right\}\right|, NP_{L3} = \frac{1}{|\mathcal{G}|} \sum_{s_i \in \mathcal{G}} \frac{1}{|s_i|} |\{F_t : c_p \in g_t\}|$$

(3)

It is worth noting that longer sequences challenge all executors in dynamic vision tasks – humans demand a higher concentration of attention, and algorithms should overcome the cumulative errors. However, existing metrics all ignore the influence of the sequence length, causing us to propose new metrics for group-level evaluation, as illustrated in Equation 4.

$$P_{L3}^w(\theta_d) = \frac{\sum_{s_i \in \mathcal{G}} |\{F_t : d_c \leq \theta_d\}|}{\sum_{s_i \in \mathcal{G}} |s_i|}, P_{L3}^w = \frac{\sum_{s_i \in \mathcal{G}} |\{F_t : d_c \leq 20\}|}{\sum_{s_i \in \mathcal{G}} |s_i|}$$

$$NP_{L3}^w(\theta_d') = \frac{\sum_{s_i \in \mathcal{G}} \left|\left\{F_t : \mathcal{N}(d_c') \leq \theta_d'\right\}\right|}{\sum_{s_i \in \mathcal{G}} |s_i|}, NP_{L3}^w = \frac{\sum_{s_i \in \mathcal{G}} |\{F_t : c_p \in g_t\}|}{\sum_{s_i \in \mathcal{G}} |s_i|}$$

(4)

## 3 EXPERIMENTS

### 3.1 A COMPREHENSIVE COMPARISON OF HUMAN-MACHINE DYNAMIC VISION ABILITY

We first interest in the comprehensive performance of human-machine DVA. Subject_Top and Subject_Bottom represent the best and worst subjects, and Subject_Mean denotes the average. Figure 3 (I) shows the distribution of $NP_{L2}$ in all sequences. As a sequence-level evaluation, box width represents stability, box position represents performance. Figure 3 (II) illustrates the normalized precision plot weighted by sequence length and the $NP_{L3}^w$ (group-level evaluation) of all executors. Clearly, humans score higher than most machines, indicating that the DVA of machines is still far from humans. Besides, the boxplot distribution shows that machines' performance is more unstable. It is worth noting that the SOTA methods have been comparable to Subject_Bottom in most cases, indicating that the research routes do significantly close the gap with humans.

As tasks become more complicated (from short-term tracking to global instance tracking), executors not only need to perceive moving targets, but also should have a good cognitive ability to quickly re-locate the target when its position suddenly changes. Figure 8 illustrates the performance of human-machine DVA under various task constraints. Based on experimental results, we find that: (1) A typical short-term tracking task only needs perceptron ability. Thus, both humans and most

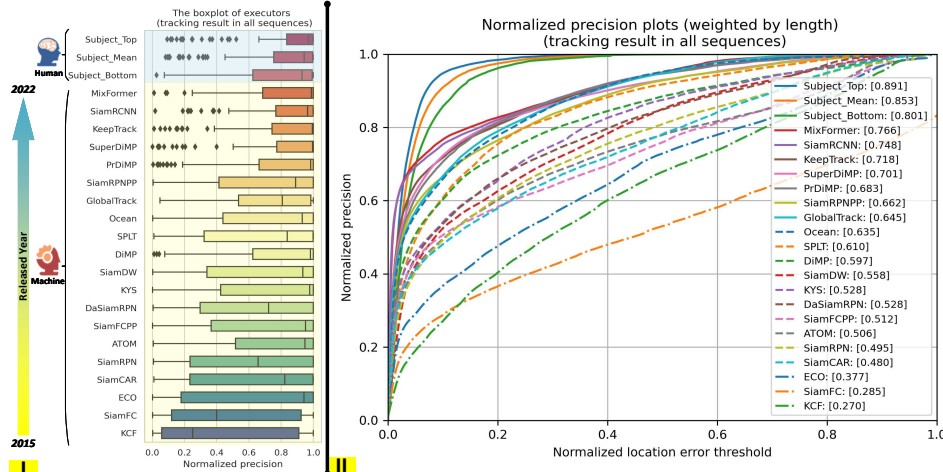

Figure 3: A comprehensive comparison of human-machine dynamic vision ability in all sequences. (I) Boxplots enumerate the distribution of $NP_{L2}$ scores. The blue area represents the maximum, mean, and minimum scores of human subjects, the yellow area represents machines. (II) The normalized precision plot weighted by sequence length and the $NP_{L3}^{w}$ scores.

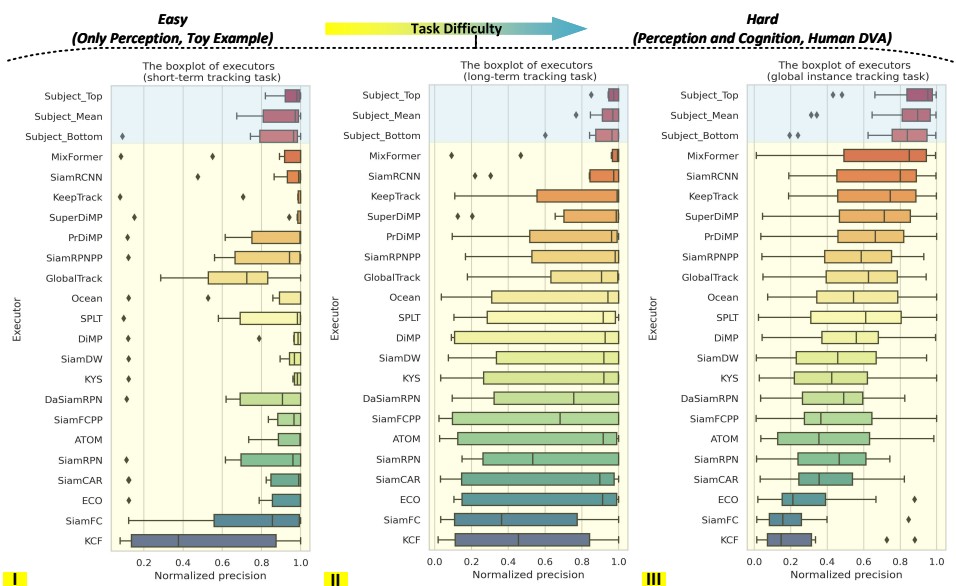

Figure 4: A comprehensive comparison of human-machine dynamic vision ability in short-term tracking (I), long-term tracking (II), and global instance tracking (III) (based on $NP_{L2}$ scores).

machines perform well in tracking the moving target; some SOTA machines even outperform humans in accuracy and stability. (2) For the long-term tracking task, since the sequence length increases and the target disappearance is allowed during the tracking process, executors should locate the target when it reappears in the frame. Results demonstrate that humans can maintain high tracking precision, but the performance of most algorithms drops significantly. (3) Global instance tracking, the most difficult task that sequence length increases with shot-cuts and scene transitions, challenges almost all algorithms. However, humans can still sustain efficient and stable tracking, indicating that the cognitive ability of humans is much better than machines.

Specifically, the top-3 machines with various model structurescan help us to understand the performance of different modeling processes (Table 2). MixFormer (Cui et al. (2022)), a simple end-to-end model based on transformer structure, performs well in both short-term and long-time tracking tasks,

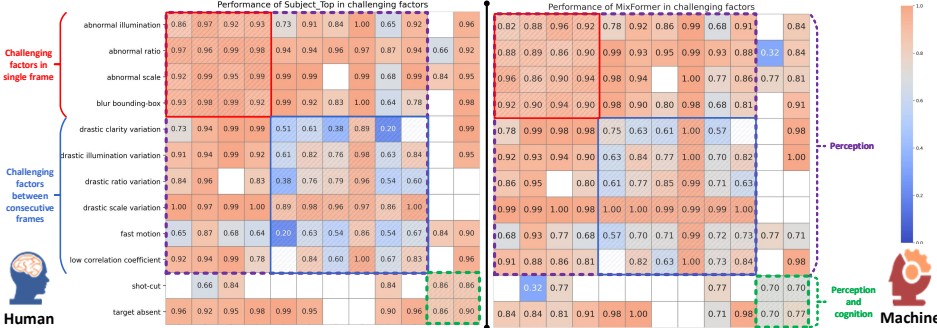

Figure 5: Performance of human and machine (MixFormer) on challenging factors. Each cell represents a mixture of challenging factors, and the blank cells represent no such combination. Cell color represents the executor's mean $NP_{L2}$ score across all videos that satisfy the combination. The purple rectangle represents short-term tracking with challenging factors (*i.e.*, target presents from beginning to end, related to perception, mainly composed in groups H to Q), of which the red and blue rectangles represent static and dynamic challenging factors separately. The green rectangle represents task constraints (*i.e.*, allowing target absence or shot switching, related to both perception and cognition, mainly composed in groups C to G). For detailed results, please refer to Appendix D.5.

indicating that the Mixed Attention Module (MAM) and a straightforward detection head can provide powerful tracking ability. The SiamRCNN (Voigtlaender et al. (2020)) model combines a two-stage scheme with a new trajectory-based dynamic planning algorithm, and uses re-detection to accomplish stable tracking. KeepTrack (Mayer et al. (2021)) model improved by SuperDiMP (Danelljan et al. (2020)) has an enhanced ability to discriminate interferers and perform well in short-time tracking, but lacking a mechanism for processing target mutation decreases its execution in longer sequences. However, all these three machines have a performance gap with humans in the global tracking task. Since MixFormer performs best in most cases, we select it as a representation to accomplish more detailed analyses in the following sections.

**Cognition.** We further explore the effect of sequence length and shot-cuts, which requires the executor to have better cognition ability to re-locate the target in complex situations. Figure 13 in Appendix shows the performance variation of human-machine under different videos. Clearly, the SOTA machine's performance fluctuates wildly when the shot-cut occurs, while the subject can still maintain a stable tracking ability. In addition, machines may ultimately lose the target after frequent shot-cuts, but humans can quickly relocate the target in a new shot. When the sequence length increases, the performance fluctuation of machines will continue to increase, but humans are better than machines in terms of fluctuation amplitude and tracking effect. It reveals that when the target motion or apparent information is mutated, the performance of machines fluctuates sharply, while humans can keep robust and precise target tracking based on their high cognition ability.

**Perceptron.** Given that multiple challenging factors may coexist in a video, we show the performance of human and SOTA machine under different factor combinations in Figure 5. Obviously, the purple rectangle is primarily related to the perceptive ability, and blue rectangle reflects the dynamic changes between consecutive frames. We can find that most cells in the blue rectangle are challenging for both algorithms and humans, but algorithms are slightly better than humans in some factors like fast motion and small targets. Conversely, task constraints represented by the green rectangle have less impact on humans, but it powerfully influences machines. This phenomenon is consistent with the above analysis of human-machine cognitive abilities.

**Further analyses about humans.** Former analyses indicate that *to err is human*. Thus, we provide detailed human performance analysis in Appendix D.6, which indicates that fast-moving targets and small targets are challenging to track, but tracking in a long sequence is not difficult for humans.

The above results show that current research, which mainly focuses on improving tracking robustness under short-term tracking tasks, has progressed. However, machines still have a gap with humans when task constraints are widened to a more general situation. Some bad cases are illustrated in Figure 14; more results based on different indicators are listed in Appendix D.3 to D.4.

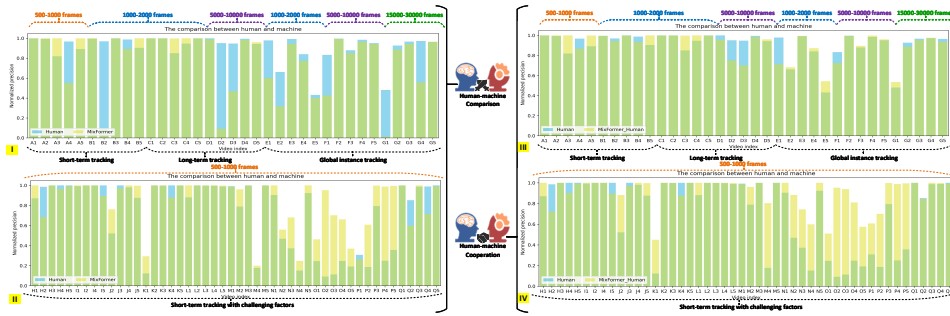

Figure 6: The comparison and cooperation of humans and machines, based on $NP_{L2}$ score. (I-II) Performance of humans (blue bar) and machines (MixFormer, yellow bar) on different sequences. (III-IV) Performance of humans (blue bar) and human-machine cooperations (MixFormer_Human, yellow bar) on different sequences. The green bar represents the overlapping part.

## 3.2  MACHINE PERFORMANCE IMPROVEMENTS WHEN COOPERATING WITH HUMANS

The analysis of Section 3.1 shows that humans and machines can adapt to different challenging factors, so an intuitive idea is *what if we allow human-machine cooperation?*

The cooperation mechanism is designed as follows: when machines fail to track, subjects can provide the current target position and restart machines from the failure frame. In this cooperation pattern, machines are responsible for performing relatively simple but repetitive tasks, while humans are responsible for supplementing information at critical moments. Specifical examples and detailed information are listed in Appendix C.7.

Figure 6 illustrates the performance of humans, machines, and human-machine collaborators. Compared with machine itself, human-machine collaborator is greatly improved and even outperforms humans in most cases. Subfigures of Figure 3 and Figure 24 indicate that human-machine collaborators have significantly improved their performance based on the assistance of humans. Therefore, humans and machines have superb possibilities for cooperation in dynamic vision tasks, which not only combines strengths and decreases weaknesses, but also saves human resources and improves efficiency. For detailed results based on different indicators, please refer to Appendix E.1 to  E.3.

## 4  CONCLUSIONS AND FUTURE WORK

In this paper, we aim to answer *how far the gap is between humans and machines in dynamic vision tasks,* and design an overall evaluation framework from three aspects – we choose 87 videos to construct the experimental environment, select 20 representative algorithms and 15 human subjects to form the executors, and quantify their DVA with a strict evaluation process. Results indicate that SOTA machines are close to the lower limit of human subjects, meaning *the DVA gap between humans and machines has significantly closed.* Besides, humans and machines excel at different dynamic vision tasks, as humans can continuously track in general environments and longer sequences based on better cognitive ability, while SOTA machines can maintain efficient and stable tracking ability in short-term tracking with excellent perceptual ability. Finally, we also find that the human-machine cooperative executor can perform even better than humans, which provides a possibility of human-machine cooperation for application scenarios like self-driving systems.

As a preliminary research of the human-machine DVA, we have obtained interesting conclusions above, but some aspects still deserve further exploration. For example, human DVA is usually decoupled into different tasks – SOT concentrates on locating an arbitrarily moving object, MOT aims to track multiple known objects simultaneously, and VOD focuses on detecting all known objects. Obviously, SOT is the basis of other dynamic vision tasks, thus we use it as a representative task for research. In the future, researchers can apply the proposed evaluation framework to multiple dynamic vision tasks and further explore the DVA of humans and machines. Besides, future work can select various subjects (*e.g.*, different age, gender, and occupation (Burg & Hulbert (1961); Melcher & Lund (1992))) to compose a more comprehensive human baseline.

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

# A TASK DESCRIPTION

## A.1 VISION TASK

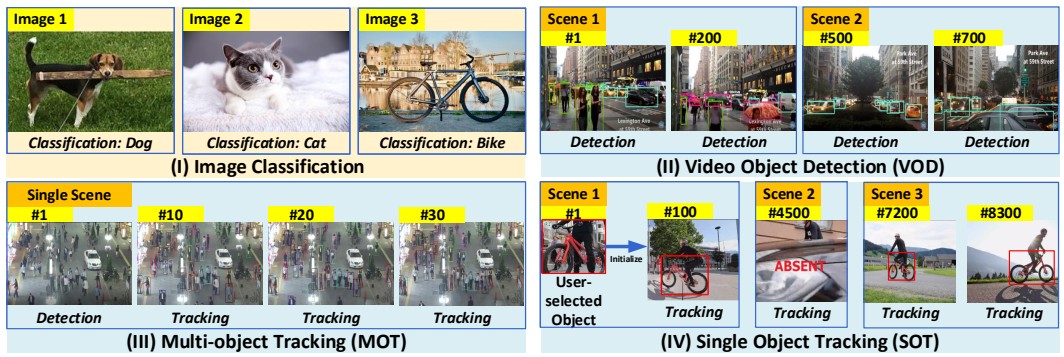

(a) The execution flow of classical computer vision tasks (Image Classification (I), Video Object Detection (II), Multi-object Tracking (III), and Single Object Tracking (IV)).

| Task | Visual Ability | Object | | Scenario | | Arbitrary Instance | Random Scenario |
|---|---|---|---|---|---|---|---|
| | | Object Category | Obejct Selection | Motion Consistency Assumption | Scene-changing | | |
| Image Classification | SVA | Limited by training set | | | | N | |
| Video Object Detection | DVA | Limited by training set | Detected | N | Y | N | Y |
| Multi-object Tracking | | Limited, usually pedestrians or vehicles | Detected in the first frame | N | N | N | N |
| Single Object Tracking | | Arbitrary | User-specified in the first frame | N | Y | Y | Y |

(b) The comparison of task characteristics.

Figure 7: The execution flow of classical computer vision tasks (a) and comparison of task characteristics (b). Obviously, SOT is closer to human dynamic vision ability.

**Image Classification.** As a static vision task, image classification (Nath et al. (2014)) allows for classifying a given image as belonging to one of the labeled and pre-defined categories. An excellent model should understand the target features and have a robust classification ability in facing challenging factors like target deformation or image blur.

**Video Object Detection (VOD).** VOD (Esteva et al. (2021)), an essential task for multiple computer vision applications, desires to accurately determine the category and location of each target in videos. However, the target category is typically limited to pre-defined categories in the training dataset.

**Multi-object Tracking (MOT).** MOT (Geuther et al. (2019)) in research usually combines with detection. MOT algorithms detect the object's position in the first frame, then calculate the similarity to determine instances belonging to the same target in consecutive frames. Thus, MOT is a model-specific task and mainly focuses on tracking specific categories (*e.g.*, pedestrians or vehicles).

**Single Object Tracking (SOT).** Unlike the above vision tasks, SOT (Wu et al. (2015)) is *category-independent*, which means it intends to track a moving target without any assumption about the target category. This characteristic allows SOT to be suitable for open-set testing with broad prospects. Besides, some recent research proposes a new task named Global Instance Tracking (GIT) (Hu et al. (2022b)) to cancel the continuous motion assumption. Thus, the SOT definition is additionally extended, making it a further step toward human dynamic vision tasks.

## A.2 SINGLE OBJECT TRACKING TASK

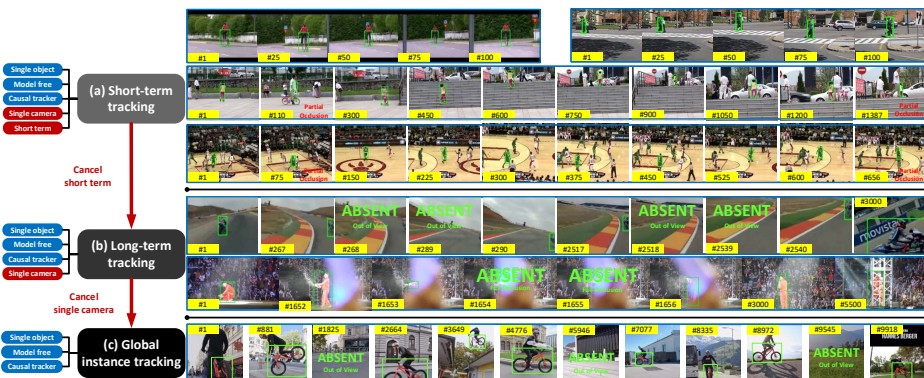

Figure 8: Representative sequence of single object tracking tasks. (a) For the short-term tracking task, the target appearance does not change significantly, but obstacles may partially obscure the target. (b) For the long-term tracking task, the target constantly moves in the same shot but may include the disappearance-reappearance process. (c) For the global instance tracking task, the target may appear in different shots and scenes, but note that its apparent information does not vary significantly.

Due to the human-like task definition of SOT, excellent task executors should not only keep tracking the moving target (*perception* in short-term tracking, long-term tracking, and global instance tracking) but also re-locate the target when its position is mutated (*cognition* in long-term tracking, and global instance tracking).

# B DATASETS AND MACHINES

## B.1 DATASETS

Table 3: The URL of open-sourced benchmarks used in this work.

| Benchmarks | URL |
|---|---|
| OTB (Wu et al. (2013; 2015)) | http://cvlab.hanyang.ac.kr/tracker_benchmark/index.html |
| VOT (Kristan et al. (2013; 2014; 2015; 2016; 2017; 2018; 2019)) | https://votchallenge.net/ |
| GOT-10k (Huang et al. (2021)) | http://got-10k.aitestunion.com |
| LaSOT (Fan et al. (2020)) | https://cis.temple.edu/lasot/ |
| VideoCube (Hu et al. (2022b)) | http://videocube.aitestunion.com |

We have correctly cited all benchmarks involved in the experiments and carefully checked the licenses involved in each benchmark (CC BY-NC-SA 4.0.). All datasets used in the experiments are open-sourced, and our usage is under the license scope.

## B.2 MACHINES

Our algorithm evaluation experiments are performed on a server with 4 NVIDIA TITAN RTX GPUs and a 64 Intel(R) Xeon(R) Gold 5218 CPU @ 2.30GHz. We use the parameters provided by the original authors.

### B.2.1 CORRELATION FILTER BASED TRACKERS

**KCF.** The CF-based trackers represented by KCF (Henriques et al. (2015)) incorporate high speed and tracking accuracy, thus becoming a representative tracking framework in the early stage. We use the Python 2.7 version based on link [1].

**ECO.** ECO (Danelljan et al. (2017)) combines CNN with CF, aiming to use deep networks to improve feature representation. The official version is released in PyTracking [2], which is a general python framework for SOT based on PyTorch. The feature representation of ECO is a combination of the first and last convolutional layer in the VGG-m network (Chatfield et al. (2014)), along with HOG (Dalal & Triggs (2005)) and Color Names (CN) (Van De Weijer et al. (2009)).

### B.2.2 SIAMESE NEURAL NETWORK BASED TRACKERS

**SiamFC.** As the originator of SNN-based trackers, SiamFC (Bertinetto et al. (2016)) [3] achieves satisfactory tracking performance by matching features between the template region and the search region through a simple network structure. The backbone of SiamFC is AlexNet (Krizhevsky et al. (2017)), which is trained on the ILSVRC15 [4] dataset for object detection in video with 50 epochs.

**SiamRPN.** SiamRPN (Li et al. (2018b)) [5] introduces the region proposal network (Girshick (2015)) to achieve accurate target regression. This model is trained based on image pairs from ImageNet-VID (Russakovsky et al. (2015)) and Youtube-BB (Real et al. (2017)) with 50 epochs.

**DaSiamRPN.** DaSiamRPN (Zhu et al. (2018)) [6] uses data augmentation to enhance the discriminative ability, which is trained based on ImageNet-VID (Russakovsky et al. (2015)), Youtube-BB (Real et al. (2017)), ImageNet (Deng et al. (2009)) and COCO (Lin et al. (2014)) with 50 epochs.

**SPLT.** SPLT (Yan et al. (2019)) [7] designs a verifier to switch global search and local search. SPLT uses MobileNet (Howard et al. (2017)) for feature extractor and downsamples the spatial resolution of the

---

[1]http://cvlab.hanyang.ac.kr/tracker_benchmark/index.html

[2]https://github.com/visionml/pytracking

[3]https://github.com/huanglianghua/siamfc-pytorch

[4]https://image-net.org/challenges/LSVRC/2015/

[5]https://github.com/huanglianghua/siamrpn-pytorch

[6]https://github.com/foolwood/DaSiamRPN

[7]https://github.com/iiau-tracker/SPLT

template feature to $1X1$ by average pooling. For the verification model, SPLT adopts ResNet50 as the backbone of verifier. The parameters of aforementioned networks are initialized with the ImageNet (Deng et al. (2009)) pre-trained models and then fine-tuned on the ImageNet-VID (Russakovsky et al. (2015)) dataset.

**SiamRPN++.** SiamRPN++ (Li et al. (2018a)) [8] introduces deeper backbone and selects ResNet (He et al. (2016)) for feature extraction. The backbone network is pre-trained on ImageNet (Deng et al. (2009)) for image labeling. Authors train the SiamRPN++ on the training sets of Youtube-BB (Real et al. (2017)), ImageNet-VID (Russakovsky et al. (2015)), and COCO (Lin et al. (2014)) and to learn a generic notion of how to measure the similarities between general objects for visual tracking.

**SiamDW.** SiamDW (Zhang & Peng (2019)) [9] selects deeper and wider backbones for feature extraction. The backbone network is pre-trained on ImageNet (Deng et al. (2009)) for image labeling. Authors train the proposed cropping inside residual (CIR) units with SiamRPN on Youtube-BB (Real et al. (2017)) and ImageNet-VID (Russakovsky et al. (2015)).

Except for backbone improvements, some SNN-based algorithms refer to detection methods to improve the network architecture.

**SiamFC++.** SiamFC++ (Xu et al. (2020)) [10] employs an anchor-free structure (Tian et al. (2019)) to eliminate the dependence on anchors. It adopts ImageNet-VID (Russakovsky et al. (2015)), COCO (Lin et al. (2014)) , Youtube-BB (Real et al. (2017)), LaSOT (Fan et al. (2020)) and GOT-10k (Huang et al. (2021)) as basic training set.

**Ocean.** Ocean (Zhang & Peng (2020)) [11] also employs the anchor-free structure. The backbone network is initialized with the parameters pretrained on ImageNet (Deng et al. (2009)). The Ocean tracker is trained on the datasets of Youtube-BB (Real et al. (2017)), ImageNet-VID (Russakovsky et al. (2015)), GOT-10k (Huang et al. (2021)) and COCO (Lin et al. (2014)).

**SiamCAR.** As another anchor-free tracker, SiamCAR (Guo et al. (2020)) [12] is trained on COCO (Lin et al. (2014)), ImageNet (Deng et al. (2009)), ImageNet-VID (Russakovsky et al. (2015)) and Youtube-BB (Real et al. (2017)).

**SiamRCNN.** SiamRCNN (Voigtlaender et al. (2020)) [13] utilizes re-detection mechanism and tracklet dynamic programming algorithm to process object disappearance. SiamRCNN is built upon the FasterRCNN (Ren et al. (2015)) implementation with a ResNet-101-FPN backbone. The backbone has been pre-trained on COCO (Lin et al. (2014)), and the overall model is trained on ImageNet-VID (Russakovsky et al. (2015)), YouTube-VOS (Xu et al. (2018)), GOT-10k (Huang et al. (2021)) and LaSOT (Fan et al. (2020)).

### B.2.3 COMBINE CORRELATION FILTER AND SIAMESE NEURAL NETWORK

**ATOM.** ATOM (Danelljan et al. (2018)) [14] tries to combine CF and SNN, and proposes a new framework to use the advantages of offline training and online updating. The authors use the training splits of LaSOT (Fan et al. (2020)) and TrackingNet (Müller et al. (2018)), and augment the training data with synthetic image pairs from COCO (Lin et al. (2014)).

**DiMP.** Based on framework proposed by ATOM, DiMP (Bhat et al. (2019)) [15] optimizes the loss function for stronger discriminative ability. The backbone network is initialized with the ImageNet (Deng et al. (2009)) weights. DIMP is trained on TrackingNet (Müller et al. (2018)), LaSOT (Fan et al. (2020)), GOT-10k (Huang et al. (2021)) and COCO (Lin et al. (2014)) datasets with 50 epochs by sampling 20,000 videos per epoch.

---

[8]https://github.com/PengBoXiangShang/SiamRPN_plus_plus_PyTorch

[9]https://github.com/researchmm/TracKit

[10]https://github.com/MegviiDetection/video_analyst

[11]https://github.com/researchmm/TracKit

[12]https://github.com/ohhhyeahhh/SiamCAR

[13]https://github.com/VisualComputingInstitute/SiamR-CNN

[14]https://github.com/visionml/pytracking

[15]https://github.com/visionml/pytracking

**PrDiMP and SuperDiMP.** PrDiMP and SuperDiMP (Danelljan et al. (2020)) [16] use probabilistic regression to improve the accuracy. The training splits of the LaSOT (Fan et al. (2020)), GOT-10k (Huang et al. (2021)), TrackingNet (Müller et al. (2018)) and COCO (Lin et al. (2014)) are used, running 50 epochs with 1000 iterations each.

**KeepTrack.** KeepTrack (Mayer et al. (2021)) [17] combines the SuperDiMP with a target candidate association network to accomplish robust tracking. The authors retrain the target candidate association network on hard sequences mined from LaSOT (Fan et al. (2020)).

### B.2.4   Custom Networks

Some other works design custom networks to solve specific problems like target absence or similar instance interference, naturally demonstrating a development from perceptual to cognitive intelligence.

**GlobalTrack.** GlobalTrack (Huang et al. (2019)) [18] does not assume motion consistency and performs a full-image search to eliminate cumulative error. The authors use COCO (Lin et al. (2014)), GOT-10k (Huang et al. (2021)) and LaSOT (Fan et al. (2020)) for model training.

**KYS.** KYS (Bhat et al. (2020)) [19] represents scene information as state vectors and combines them with the appearance model to locate the object. The authors use the training splits of TrackingNet (Müller et al. (2018)), GOT-10k (Huang et al. (2021)) and LaSOT (Fan et al. (2020)) for model training.

**MixFormer.** MixFormer (Cui et al. (2022)) [20] designs an end-to-end transformer-based framework to simultaneously accomplish feature extraction and target information integration. The authors use COCO (Lin et al. (2014)), TrackingNet (Müller et al. (2018)), GOT-10k (Huang et al. (2021)) and LaSOT (Fan et al. (2020)) for model training.

---

[16]https://github.com/visionml/pytracking
[17]https://github.com/visionml/pytracking
[18]https://github.com/huanglianghua/GlobalTrack
[19]https://github.com/visionml/pytracking
[20]https://github.com/MCG-NJU/MixFormer

# C EXPERIMENT ORGANIZATION

## C.1 EXPERIMENTAL ENVIRONMENT

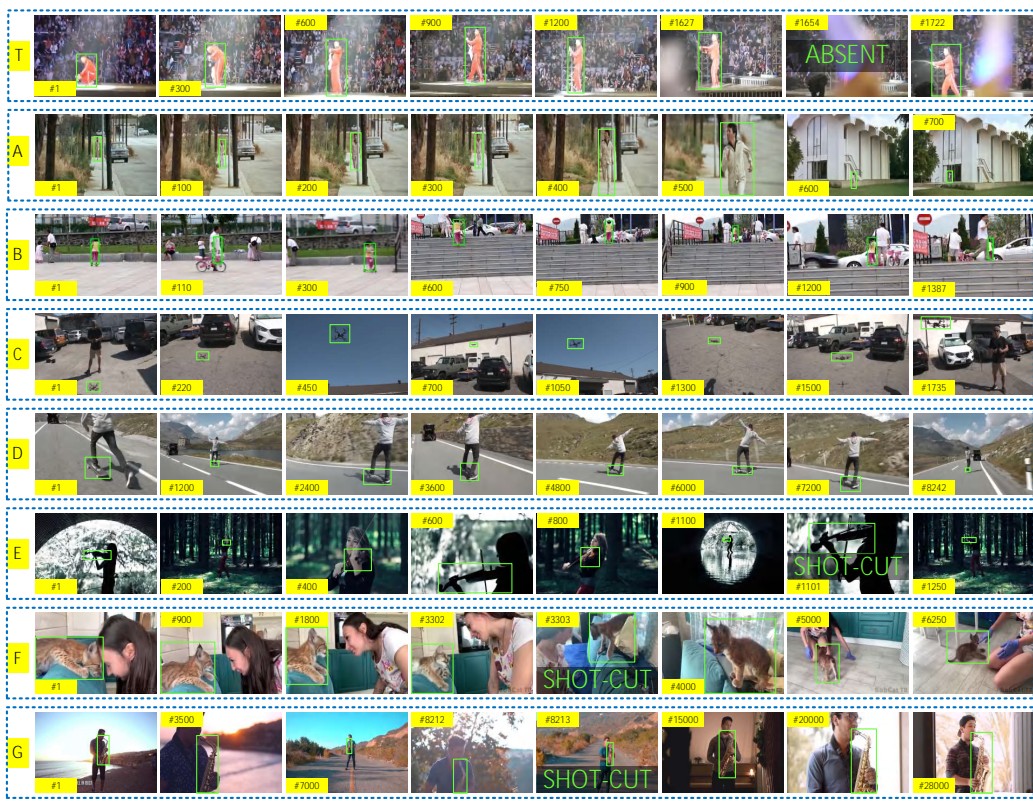

Figure 9: Representative sequences used in experiments. (T) Example of TEST videos, which help the experimenter familiar with the operation. (A) Example of short-term tracking videos, the sequence length of this group is concentrated in 500-1000 frames. (B) Example of short-term tracking videos, the sequence length of this group is concentrated in 1000-2000 frames. (C) Example of long-term tracking videos, the sequence length of this group is concentrated in 1000-2000 frames. (D) Example of long-term tracking videos, the sequence length of this group is concentrated in 5000-10000 frames. (E) Example of global instance tracking videos, the sequence length of this group is concentrated in 1000-2000 frames. (F) Example of global instance tracking videos, the sequence length of this group is concentrated in 5000-10000 frames. (G) Example of global instance tracking videos, the sequence length of this group is concentrated in 15000-30000 frames.

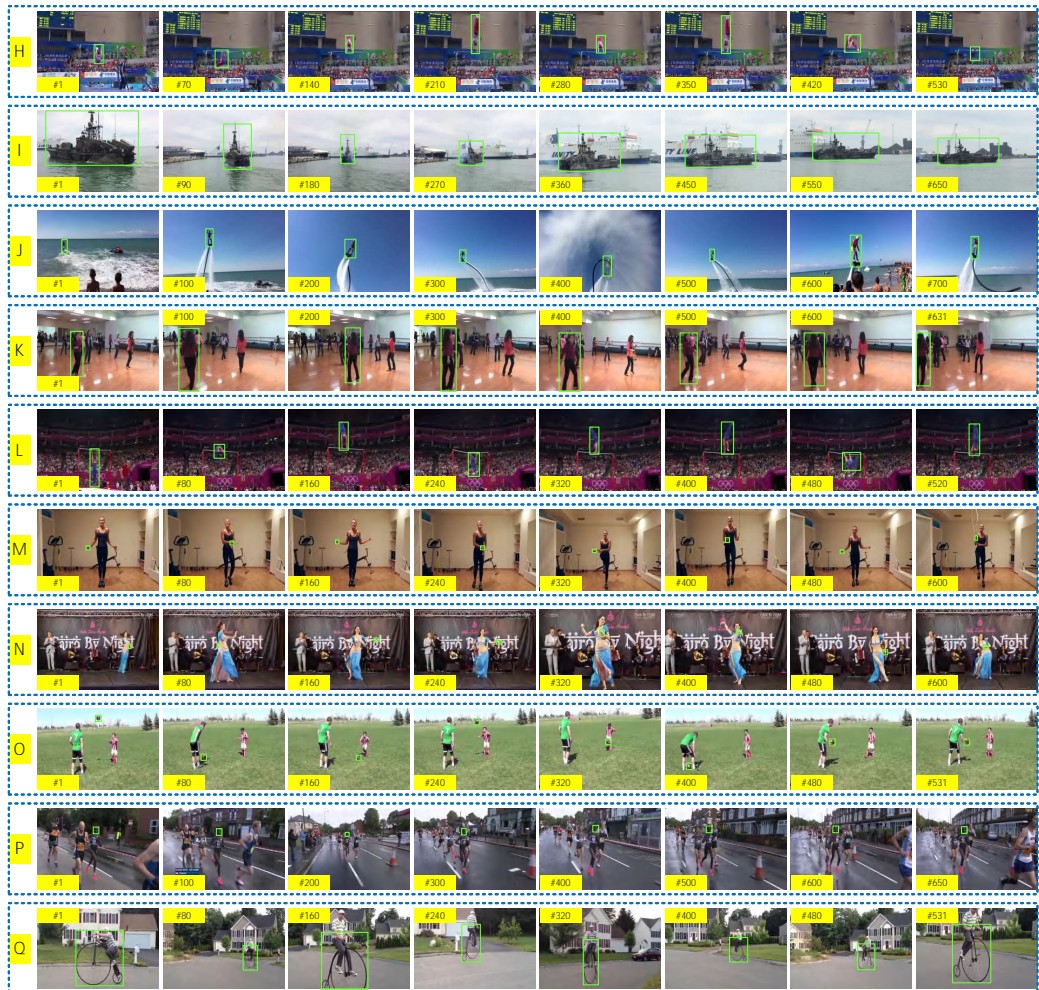

Figure 10: Representative sequences used in experiments, the sequence length in this part is concentrated in 500-1000 frames. (H) Example of tracking the target with abnormal ratio. (I) Example of tracking the target with abnormal scale. (J) Example of tracking the target with abnormal illumination. (K) Example of tracking the target with blur bounding-box. (L) Example of tracking the target with drastic ratio variation. (M) Example of tracking the target with drastic scale variation. (N) Example of tracking the target with drastic illumination variation. (O) Example of tracking the target with drastic clarity variation. (P) Example of tracking the target with fast motion. (Q) Example of tracking the target with low correlation coefficient.

C.2 EXPERIMENTAL STATEMENT

We have obtained the approvals of all experiments, the template is provided by our IRB. Every experiment has signed an experimental statement before the experiment. Here we list the contents:

**Purpose** *You have been asked to participate in a research study that studies the dynamic visual abilities of humans. We would like your permission to enroll you as a participant in this research study. The instruments involved in the experiment are a computer screen and a mouse. The experimental task consisted of gazing at the presented image sequence and manipulating the mouse to point as consistently as possible at the center of the moving target. You will be given specific instructions for the task before it begins.*

**Procedure** *In this study, you should read the experimental instructions and ensure that you understand the experimental content. The whole experiment process lasts about one hour, and the experiment is divided into the following steps:*

- *Read and sign the experimental statement;*
- *Test the experimental instrument, and adjust the seat height, sitting posture, and the distance between your eyes and the screen. Please ensure that you are in a comfortable sitting position during the experiment;*
- *Two TEST videos will be played. You should comprehend the specific instrument operation rules and be familiar with the experimental process through the test videos;*
- *Start the FORMAL experiment. Please follow the operation to watch the test sequence and complete the relevant operations. Note that after watching each video sequence, you should have a rest;*
- *After the experiment, you need to fill in a questionnaire.*

**Risks and Discomforts** *The only potential risk factor for this experiment is trace electron radiation from the computer. Relevant studies have shown that radiation from computers and related peripherals will not cause harm to the human body. To rule out the impact of COVID-19 on the experiment, all participants completed nucleic acid tests before the experiment to ensure their health. Everyone needs to wear a mask throughout the experiment. In addition, the relevant devices are cleaned and disinfected during the experiment.*

**Costs** *Each participant who completes the experiment will be paid 200 RMB.*

**Confidentiality** *The results of this study may be published in an academic journal/book or used for teaching purposes. However, your name or other identifiers will not be used in any publication or teaching materials without your specific permission. In addition, if photographs, audio tapes or videotapes were taken during the study that would identify you, then you must give special permission for their use.*

*I confirm that the purpose of the research, the study procedures and the possible risks and discomforts as well as potential benefits that I may experience have been explained to me. All my questions have been satisfactorily answered. I have read this consent form. My signature below indicates my willingness to participate in this study.*

## C.3  DEVICE SELECTION

How to measure the DVA of humans is an important section of our work. Previous works mainly use eye-tracker (Hu et al. (2022b); Xia et al. (2021)) or mouse (Geirhos et al. (2018; 2020; 2021)) to record human visual ability. To select a better measurement method, we divide them into active (mouse) and passive (eye-tracker) methods, then compare them from practical and theoretical views:

- From a practical view, we have organized all subjects to experience both mouse and eye-tracker, and they all express that the user experience of the mouse is better. A possible reason is that the eye-tracker passively records eye movements, causing the effect to be limited by many factors (*e.g.*, the device accuracy, the user's posture, the relative distance between eyes and the eye-tracker, and the user's glasses). Besides, many subjects mention that the eye-tracker can only effectively track eye movements for a short period. When the experimental time is prolonged, the recording result easily shifts, while subjects can only adjust their posture rather than actively correct the capture result. Conversely, using a mouse can minimize the above problems. Although mouse may also have some limitations, its accuracy is greater than eye-tracker in our experiments. For example, when subjects notice that the mouse position is far from the target, they can actively move the mouse and immediately correct the record result.

- From a theoretical view, some researchers (Holmqvist et al. (2022)) also indicate that the eye-tracker is affected by the tracking ratio (*i.e.*, the amount of eye-tracking data lost). Blinks or brief vision drifts can cause a lower tracking ratio, which requires the subject to keep a high degree of concentration when using eye-trackers. To prove this view, we check recent articles and find that experiments based on eye-trackers are either for observing the images (subjects have sufficient time to adjust the position per image) (Xia et al. (2020)) or watching a small amount of video (*e.g.*, 6 videos) (Hu et al. (2022b)). Nevertheless, our experiment requests each participant to watch 17 videos (including long videos). Thus, while maintaining a high concentration of attention, subjects frequently change their sitting position unconsciously, causing the eye-tracker performs poorly in our experiment. What's more, we find that previous works (Geirhos et al. (2018; 2020; 2021)) have allowed subjects to perform image classification tasks with a mouse-based method, and a neuro training company named Reflexion also uses touch screens to provide DVA training for athletes, which means recording the observation position based on hand operation has been applied in academia and industry. Thus, we finally select the mouse-based experimental method.

## C.4 EXPERIMENT PROCESS

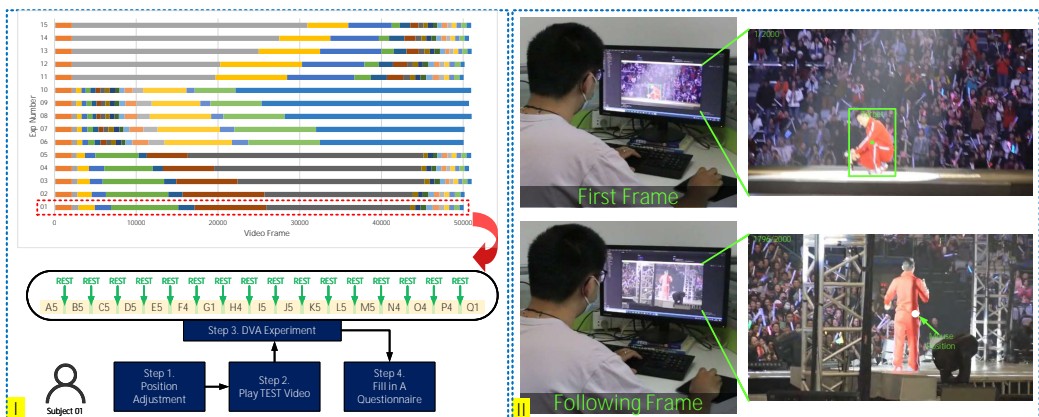

Figure 11: Process of DVA experiment. (I) 15 subjects participated in the DVA experiment. For each video number, the first letter represents the video group, and the second letter represents the index. Each group contains 5 videos (*e.g.*, A1-A5), and each video is observed by 3 subjects (*e.g.*, A1 is observed by subjects 05, 10, 15, respectively). Subjects whose serial numbers differ by 5 should watch the same content but in a different order. (II) The experiment includes two steps: observing and remembering the target features in the first frame, then moving the mouse to track the target in subsequent frames. Note that subject has a rest time to relieve visual fatigue between two videos.

## C.5 QUESTIONNAIRE

Subjects should fill in a questionnaire to evaluate their performance and make recommendations. Here we list the contents:

*This questionnaire aims to investigate subjects' opinions and self-evaluation.*

**Q1. Your name:**

**Q2. Your ID (e.g., subject 1):**

**Q3. Your age:**

**Q4. Your vision condition (fill in the left and right eyes separately):**

**Q5. Please rank the following factors in order of perceived difficulty during your experiment (order from difficult to easy):**

- *Video sequence is too long;*
- *Frequent shot changes;*
- *Screen shaking;*
- *Target scale is too small;*
- *Similar objects interference;*
- *Target moves fast;*
- *Target is occluded.*

**Q6. In addition to the challenging factors in the previous question, what other factors may influence your observation?**

**Q7. Please provide your self-evaluation score (0 to 100) in short sequences (1-3 minutes):**

**Q8. Please provide your self-evaluation score (0 to 100) in medium sequences (3-10 minutes):**

**Q9. Please provide your self-evaluation score (0 to 100) in long sequences (more than 10 minutes):**

**Q10. What was the most impressive sequence in the experiment? Why are you impressed by it? (e.g., the video is too long, the content of the video is exceptional, the video has some challenging factors that lead to poor performance, etc.)**

**Q11. Do you have any suggestions or comments for this experiment?**

## C.6 GROUP INFORMATION

Table 4: Groups of subjects and corresponding videos. Subjects indicated by the same color watch the same content but in a different order.

(a) Each human subject will watch 19 videos, including 2 TEST videos and 17 FORMAL experiment videos. For each video number, the first letter represents the video group, and the second letter represents the number of videos in that group. Except for the TEST videos, each group contains five videos (e.g., A1-A5), and each video is observed by three subjects (e.g., A1 is observed by subjects 05, 10, and 15, respectively).

| Group | I | | | | | II | | | | | III | | | | |
|---|---|---|---|---|---|---|---|---|---|---|---|---|---|---|---|
| Exp | 01 | 02 | 03 | 04 | 05 | 06 | 07 | 08 | 09 | 10 | 11 | 12 | 13 | 14 | 15 |
| Video | T1 | T1 | T1 | T1 | T1 | T1 | T1 | T1 | T1 | T1 | T1 | T1 | T1 | T1 | T1 |
| | T2 | T2 | T2 | T2 | T2 | T2 | T2 | T2 | T2 | T2 | T2 | T2 | T2 | T2 | T2 |
| | A5 | A2 | A3 | A4 | A1 | H4 | H5 | H3 | H2 | H1 | G1 | G2 | G3 | G4 | G5 |
| | B5 | B4 | B3 | B2 | B1 | I5 | I1 | I2 | I3 | I4 | F4 | F5 | F3 | F2 | F1 |
| | C5 | C3 | C2 | C4 | C1 | J5 | J4 | J3 | J2 | J1 | D5 | D4 | D3 | D2 | D1 |
| | D5 | D4 | D3 | D2 | D1 | K5 | K4 | K3 | K2 | K1 | E5 | E4 | E3 | E2 | E1 |
| | E5 | E4 | E3 | E2 | E1 | L5 | L4 | L1 | L2 | L3 | C5 | C3 | C2 | C4 | C1 |
| | F4 | F5 | F3 | F2 | F1 | M5 | M4 | M1 | M2 | M3 | B5 | B4 | B3 | B2 | B1 |
| | G1 | G2 | G3 | G4 | G5 | N4 | N3 | N5 | N1 | N2 | A5 | A2 | A3 | A4 | A1 |
| | H4 | H5 | H3 | H2 | H1 | O4 | O5 | O1 | O2 | O3 | H4 | H5 | H3 | H2 | H1 |
| | I5 | I1 | I2 | I3 | I4 | P4 | P5 | P1 | P2 | P3 | I5 | I1 | I2 | I3 | I4 |
| | J5 | J4 | J3 | J2 | J1 | Q1 | Q2 | Q3 | Q4 | Q5 | J5 | J4 | J3 | J2 | J1 |
| | K5 | K4 | K3 | K2 | K1 | A5 | A2 | A3 | A4 | A1 | K5 | K4 | K3 | K2 | K1 |
| | L5 | L4 | L1 | L2 | L3 | B5 | B4 | B3 | B2 | B1 | L5 | L4 | L1 | L2 | L3 |
| | M5 | M4 | M1 | M2 | M3 | C5 | C3 | C2 | C4 | C1 | M5 | M4 | M1 | M2 | M3 |
| | N4 | N3 | N5 | N1 | N2 | D5 | D4 | D3 | D2 | D1 | N4 | N3 | N5 | N1 | N2 |
| | O4 | O5 | O1 | O2 | O3 | E5 | E4 | E3 | E2 | E1 | O4 | O5 | O1 | O2 | O3 |
| | P4 | P5 | P1 | P2 | P3 | F4 | F5 | F3 | F2 | F1 | P4 | P5 | P1 | P2 | P3 |
| | Q1 | Q2 | Q3 | Q4 | Q5 | G1 | G2 | G3 | G4 | G5 | Q1 | Q2 | Q3 | Q4 | Q5 |

(b) Subjects in group I watch the video length from short to long, then become short again. Subjects in group II watch videos of increasing length. Subjects in group III watch videos of decreasing length.

| Group | I | | | | | II | | | | | III | | | | |
|---|---|---|---|---|---|---|---|---|---|---|---|---|---|---|---|
| Exp | 01 | 02 | 03 | 04 | 05 | 06 | 07 | 08 | 09 | 10 | 11 | 12 | 13 | 14 | 15 |
| Frame | 120 | 120 | 120 | 120 | 120 | 120 | 120 | 120 | 120 | 120 | 120 | 120 | 120 | 120 | 120 |
| | 2000 | 2000 | 2000 | 2000 | 2000 | 2000 | 2000 | 2000 | 2000 | 2000 | 2000 | 2000 | 2000 | 2000 | 2000 |
| | 844 | 700 | 708 | 725 | 610 | 631 | 650 | 630 | 581 | 528 | 17592 | 18166 | 22872 | 25379 | 28828 |
| | 1981 | 1741 | 1500 | 1377 | 1000 | 651 | 570 | 571 | 621 | 645 | 8750 | 10000 | 7500 | 6250 | 5000 |
| | 1961 | 1735 | 1568 | 1865 | 1266 | 701 | 691 | 681 | 600 | 531 | 8242 | 7628 | 7509 | 5933 | 5307 |
| | 8242 | 7628 | 7509 | 5933 | 5307 | 806 | 671 | 631 | 631 | 551 | 2000 | 1750 | 1500 | 1250 | 1000 |
| | 2000 | 1750 | 1500 | 1250 | 1000 | 741 | 688 | 520 | 598 | 671 | 1961 | 1735 | 1568 | 1865 | 1266 |
| | 8750 | 10000 | 7500 | 6250 | 5000 | 659 | 601 | 510 | 531 | 550 | 1981 | 1741 | 1500 | 1377 | 1000 |
| | 17592 | 18166 | 22872 | 25379 | 28828 | 601 | 601 | 585 | 556 | 571 | 844 | 700 | 708 | 725 | 610 |
| | 631 | 650 | 630 | 581 | 528 | 601 | 671 | 501 | 531 | 581 | 631 | 650 | 630 | 581 | 528 |
| | 651 | 570 | 571 | 621 | 645 | 670 | 681 | 601 | 620 | 650 | 651 | 570 | 571 | 621 | 645 |
| | 701 | 691 | 681 | 600 | 531 | 501 | 501 | 531 | 542 | 561 | 701 | 691 | 681 | 600 | 531 |
| | 806 | 671 | 631 | 631 | 551 | 844 | 700 | 708 | 725 | 610 | 806 | 671 | 631 | 631 | 551 |
| | 741 | 688 | 520 | 598 | 671 | 1981 | 1741 | 1500 | 1377 | 1000 | 741 | 688 | 520 | 598 | 671 |
| | 659 | 601 | 510 | 531 | 550 | 1961 | 1735 | 1568 | 1865 | 1266 | 659 | 601 | 510 | 531 | 550 |
| | 601 | 601 | 585 | 556 | 571 | 8242 | 7628 | 7509 | 5933 | 5307 | 601 | 601 | 585 | 556 | 571 |
| | 601 | 671 | 501 | 531 | 581 | 2000 | 1750 | 1500 | 1250 | 1000 | 601 | 671 | 501 | 531 | 581 |
| | 670 | 681 | 601 | 620 | 650 | 8750 | 10000 | 7500 | 6250 | 5000 | 670 | 681 | 601 | 620 | 650 |
| | 501 | 501 | 531 | 542 | 561 | 17592 | 18166 | 22872 | 25379 | 28828 | 501 | 501 | 531 | 542 | 561 |
| Total | 50052 | 50165 | 51038 | 50710 | 50970 | 50052 | 50165 | 51038 | 50710 | 50970 | 50052 | 50165 | 51038 | 50710 | 50970 |

## C.7 COOPERATION MECHANISM

### C.7.1 DETAILED INFORMATION ABOUT COOPERATION MECHANISM

---

**Algorithm 1** Framework of cooperation mechanism

---

**Input:** $\mathbf{T}$: tracker, which has two functions (INITIALIZATION and TRACK);
$\mathcal{S} = \{F_1, F_2, \ldots, F_t, \ldots\}$: original sequence;
$\mathcal{G} = \{g_1, g_2, \ldots, g_t, \ldots\}$: the set of ground-truth, $g_t = (x_{gt}, y_{gt}, w_{gt}, h_{gt})$ represents the ground-truth bounding box in $F_t$;
$\mathcal{B} = \{b_1, b_2, \ldots, b_t, \ldots\}$: the set of blur degree, $b_t$ represents the blur degree value of $F_t$, higher value means more clarity;
$\mathcal{R} = \{r_1, r_2, \ldots, r_t, \ldots\}$: the set of object relative scale, $r_t$ represents the object relative scale value of $F_t$, small value means tiny object
**Output:** $\mathcal{P} = \{p_1, p_2, \ldots, p_t, \ldots\}$: the set of tracking results, $p_t = (x_{pt}, y_{pt}, w_{pt}, h_{pt})$ represents the predicted bounding box in $F_t$;
$\mathcal{I}$: the set of cooperation frames, $|\mathcal{I}|$ can be regarded as number of cooperations

```
/* Step 1:   find suitable cooperation frames                */
```
1  set candidate cooperation frames $\mathcal{C} = \phi$
   **for** $i \leftarrow 0$ **to** $|\mathcal{S}| - 1$ **do**
2       **while** $(b_i > \text{MEDIAN}(\mathcal{B})) \wedge (r_i > \text{MEDIAN}(\mathcal{R}))$ **do**
3          $\mathcal{C} \leftarrow \mathcal{C} \cup i$

```
/* Step 2:   human-machine cooperation                       */
```
4  set tracking results $\mathcal{P} = \phi$
   set initialization locations $\mathcal{I} = \phi$
   set failure counter $\alpha = 0$
   **for** $i \leftarrow 1$ **to** $|\mathcal{S}|$ **do**
5       **if** $i == 1$ **then**
6          $p_1 = \mathbf{T}.\text{INITIALIZATION}(F_1, g_1)$
           $\mathcal{P} \leftarrow \mathcal{P} \cup p_1$
           **continue**
7       **if** $(\alpha \geq 10) \wedge (i \in \mathcal{C})$ **then**
8          $p_i = \mathbf{T}.\text{INITIALIZATION}(F_i, g_i^c)$
           $\mathcal{P} \leftarrow \mathcal{P} \cup p_i$
           $\mathcal{I} \leftarrow \mathcal{I} \cup i$
           $\alpha \leftarrow 0$
           **continue**
9       **else**
10         $p_i = \mathbf{T}.\text{TRACK}(F_i)$
           $\mathcal{P} \leftarrow \mathcal{P} \cup p_i$
           $s_i = \Omega(p_i, g_i) = \frac{p_i \bigcap g_i}{p_i \bigcup g_i}$
           **if** $s_i < 0.5$ **then**
11              $\alpha \leftarrow \alpha + 1$
12            **else**
13              $\alpha \leftarrow 0$

14 **return** $\mathcal{P}, \mathcal{I}$

---

As we have metioned in Section 3.2, the cooperation mechanism is designed as follows: when machines fail to track, subjects can provide the current target position and restart machines from the failure frame. It is worth noting that human-computer cooperation is very complex, and our experiments only explore the possibility of cooperation through a simple and intuitive mechanism. There is still much space for further optimization.

Based on Algorithm 1, the human-machine cooperation is divided into the following steps. (1) **Find suitable cooperation frames.** The quality of the cooperation frame is important for tracking in the following frames. Frames with tiny objects or motion blur may decrease appearance and motion information. Thus, we first calculate the blur degree and relative target scale of all frames in $\mathcal{S}$,

and select frames that include the target with relatively clear and appropriate scale as candidate cooperation frames (please refer to Hu et al. (2022a) for more information), and generate the set of candidate cooperation position $\mathcal{C}$. (2) **Human-machine cooperation.** We use the IoU value $s_i$ (Line 10 in Algorithm 1) to determine tracking failure and consider that after ten consecutive frames of tracking failure (failure counter $\alpha \geq 10$), the algorithm will cooperate with the human at the nearest candidate cooperation frame and reinitialize (Line 8 in Algorithm 1, $g_i^c$ represents the object information provided by human). Besides, considering that many SOTA machines may re-locate the target in several following frames, we assume that $s_i \geq 0.5$ means successfully re-locate the target, and zeroing the failure counter $\alpha$ if the re-location occurs in 10 frames (Line 13 in Algorithm 1).

It is worth noting that our human-machine cooperation experiments are a preliminary exploration, and we hope to find that cooperation is possible for all algorithms. However, for a combination of 20 algorithms and 87 videos (1740 sets), it is time-consuming to arrange human subjects to participate in 1740 cooperation experiments. Therefore, we used the following strategy to organize the experiments. (1) We first use the ground-truth as target information provided to the algorithm at cooperation frames (Line 8 in Algorithm 1, here we use $g_i$ to re-initialize the algorithm). Since the ground-truth of sequences are provided by professional human annotators, which can be regarded as a representative of the highest level of human in static frames. Then we record all the cooperation frames as $\mathcal{I}_g$ ($g$ means this set is generated by ground-truth information). (2) We show the frames in $\mathcal{I}_g$ to human subjects, asking them to find the target in the current frame and annotate it with a bounding box, then we generate the human annotations $\mathcal{I}_h$ (the length of $\mathcal{I}_h$ is equal to $\mathcal{I}_g$). (3) We evaluate $\mathcal{I}_h$ and $\mathcal{I}_g$ based on $NP_{L2}$ score. **Situation 1:** If the score is higher than 0.95, we consider the human subject to have the same performance as the original data annotator. Thus, we no longer organize the cooperation experiment for this algorithm-sequence combination. **Situation 2:** On the contrary, if the score is below 0.95, we consider that the human subject has a gap with the ground-truth. We will arrange for the human subject and the algorithm to track the sequence together from the beginning, ask the human subject to annotate the target position by bounding-box in cooperation frames, and then record the collaborator's performance (Line 8 in Algorithm 1, here we use $g_i^c$ to re-initialize the algorithm).

We find that most sequences satisfied Situation 1. As the example in Figure 12, the human subject watches the first frame and clearly understands that he should locate the blue-clothed player. Then he can identify the target at frames #1125 and #6353, even if he does not watch the middle frames. This phenomenon also demonstrates the solid cognitive and memory abilities of humans.

### C.7.2 AN EXAMPLE

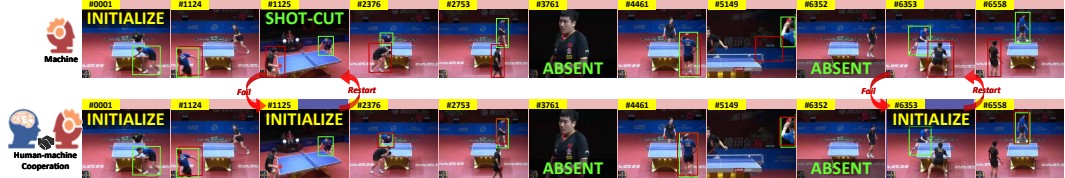

Figure 12: Example of the cooperation mechanism. The top area illustrates machine tracking results on sequence G4, and the bottom is the cooperation results. When machines fail to track (*e.g.*, #1125 and #6353, always due to target absent or shot-cut), subjects can provide the current target position and re-initialize machines to keep tracking from the failure frame.

To better illustrate the cooperation process, we use video G4 (a long video with 25,397 frames and 450 shot-cuts, describing a table tennis match with two players) as an example. The best human subject scores 0.964 ($NP_{L2}$ score) since it is easy to locate a player consistently. However, when the shot is switched, the position variation of the target player (blue-clothed player) will challenge the SOTA algorithm. Thus, it quickly drifts to the interferer (black-clothed player) and keeps wrong tracking, causing the score to decrease to 0.558. But if we allow algorithms to cooperate with humans, when the human subject finds that the algorithm fails (such as drifting to the distractor), the subject will provide correct target position to the algorithm to avoid persistent mistracking. For our experiment in G4, the human subject provides target location information 23 times while the algorithm tracks the remaining 25,374 frames. Although humans only need to participate 0.1% (23/25,397) in cooperation, the human-machine collaborator MixFormer_Human can score 0.977, indicating that cooperation can significantly improve the algorithm's tracking performance. This result is also in line with the original intention: algorithms accomplish many simple tasks, while humans focus on the key points.

# D  PERFORMANCE OF HUMANS AND MACHINES

## D.1  RESULTS ON DIFFERENT SEQUENCE LENGTHS

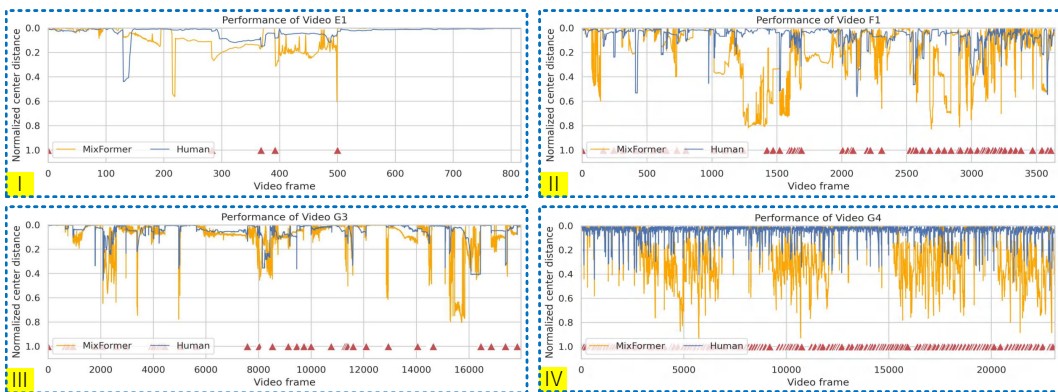

Figure 13: Performance of human-machine on different sequence lengths (the vertical coordinate indicates $NP_{L1}$ score per frame). The yellow curve represents the SOTA machine (MixFormer (Cui et al. (2022))), the blue curve represents the subject, and the red triangle represents the shot-cut. Subfigures (I-IV) represent four videos with different combinations of lengths and shots. Compared with the SOTA machine, humans can still maintain adequate DVA in the face of lengthy sequences and frequent shot switching.

## D.2   EXAMPLES ON DIFFERENT SEQUENCES

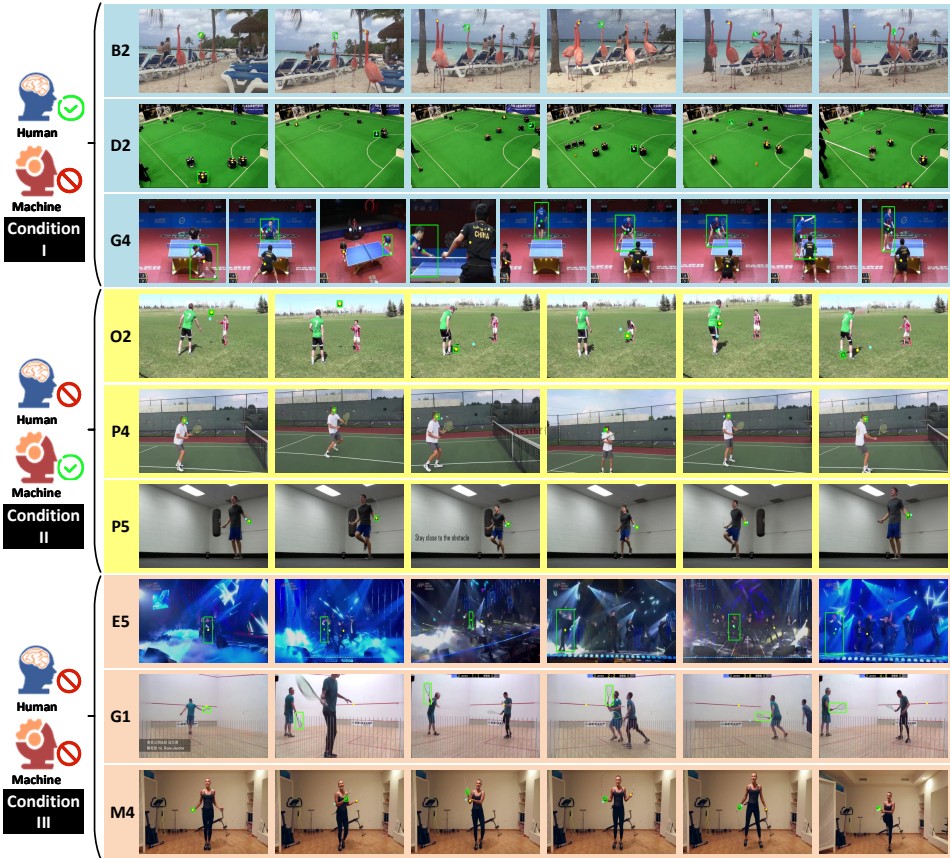

Figure 14: Schematic diagram of human-machine tracking results on partial sequences. Condition I (B2, D2, G4) represents that humans are better than machines, condition II (O2, P4, P5) indicates that machines are better than humans, and condition III (E5, G1, M4) indicates that both perform poorly. For each frame, the green rectangle and green point represent the ground-truth, the blue point represents the localization result of humans, and the yellow point represents the tracking result of machines.

Obviously, humans outperform machines in longer sequences with shot-cuts and can better distinguish the target from interfering objects (condition I). However, when facing challenges such as tracking small targets with fast movements (condition II), limited by hand-brain coordination and mouse movement speed, humans cannot precisely locate such targets at 25FPS. Note that both humans and machines may fail when multiple challenges are superimposed in one sequence (condition III).

### D.3 BOXPLOTS

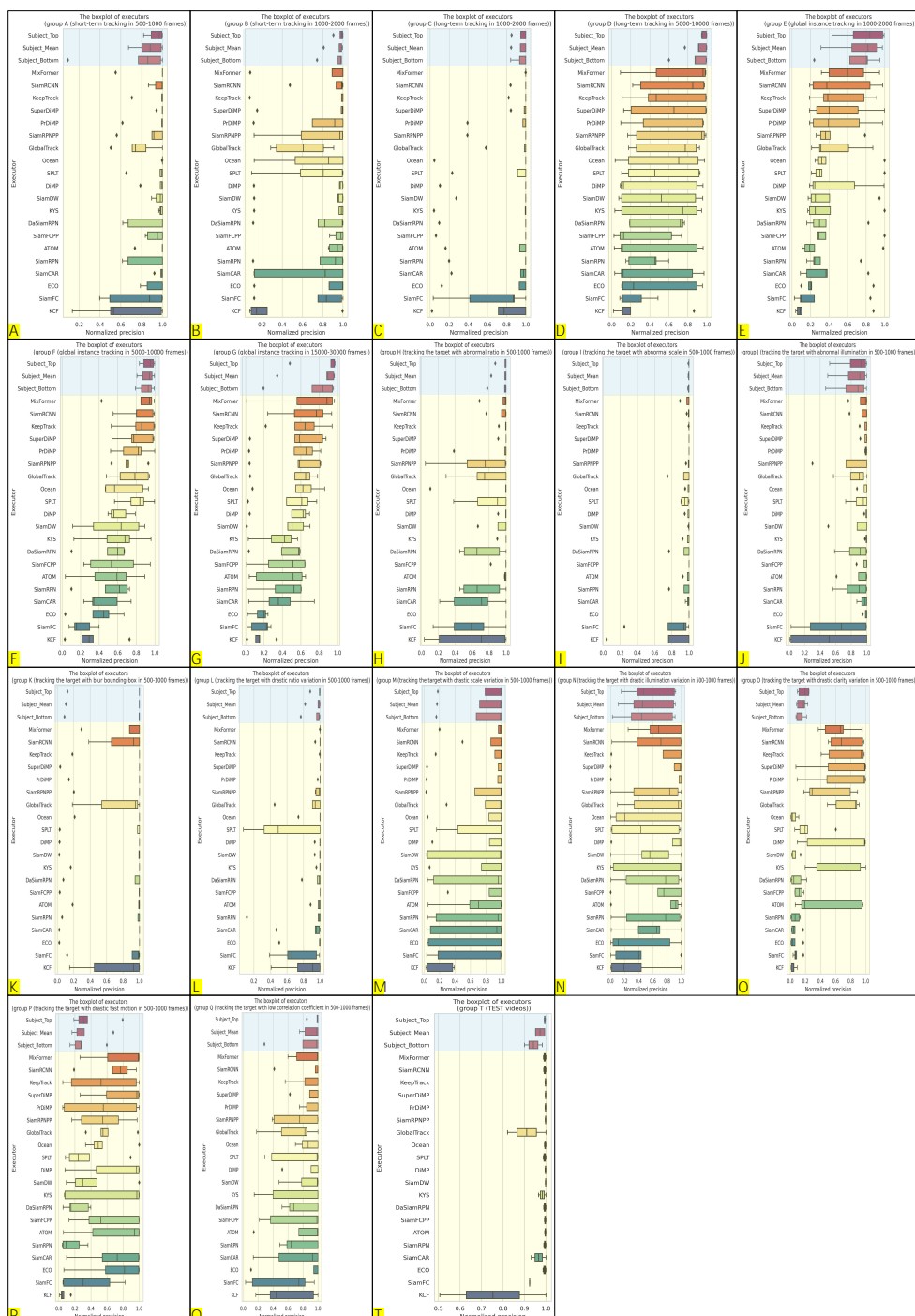

Figure 15: Boxplots enumerate the distribution of $NP_{L2}$ scores for TEST videos and Group A to Q. The blue area represents the maximum, mean, and minimum scores of human subjects, and the yellow area represents machines.

## D.4 PRECISION PLOTS

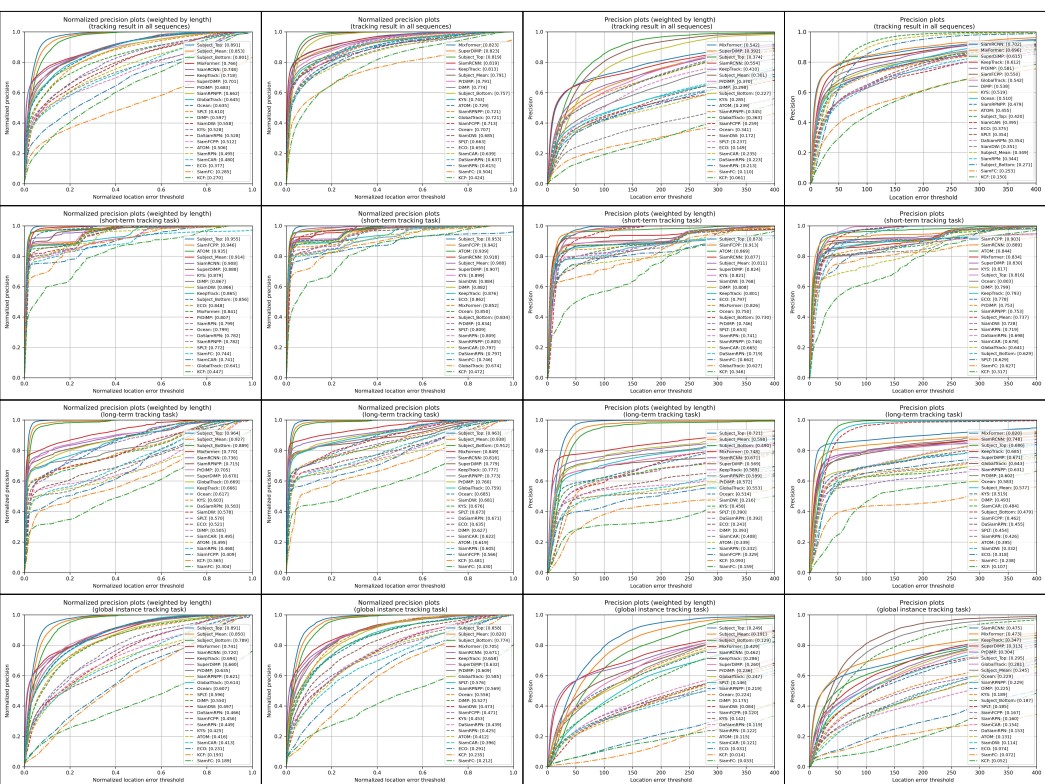

Figure 16: Results for all tasks, short-term tracking task, long-term tracking task, and global instance tracking task based on various indicators (from left to right: weighted-N-PRE plots $NP_{L3}^w(\theta_d{}')$, ranked by $NP_{L3}^w$; N-PRE plots $NP_{L3}(\theta_d{}')$, ranked by $NP_{L3}$; weighted-PRE plots $P_{L3}^w(\theta_d)$, ranked by $P_{L3}^w$; and PRE plots $P_{L3}(\theta_d)$, ranked by $P_{L3}$).

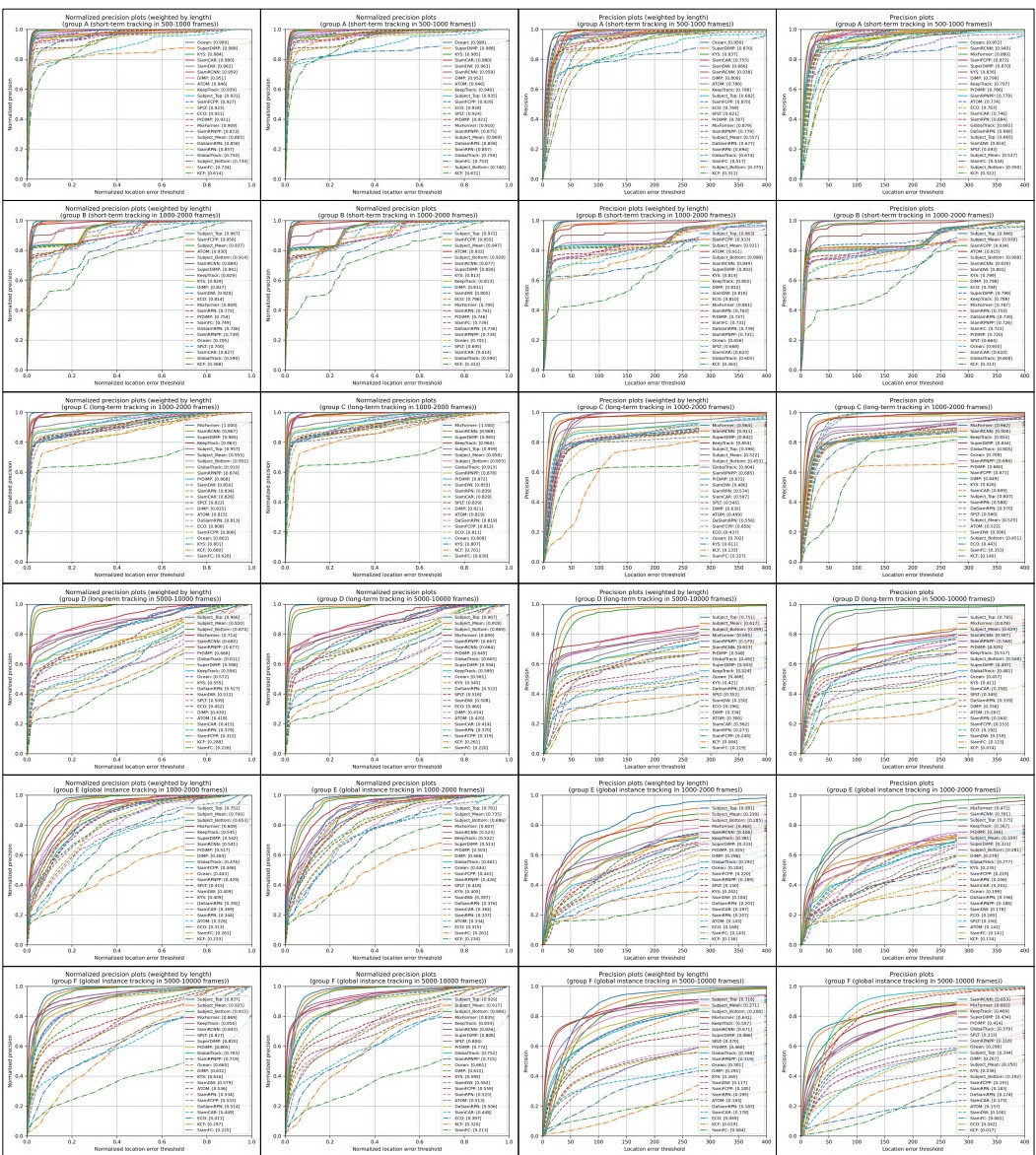

Figure 17: Results for Group A to F based on various indicators (from left to right: weighted-N-PRE plots $NP^w_{L3}(\theta_d{}')$, ranked by $NP^w_{L3}$; N-PRE plots $NP_{L3}(\theta_d{}')$, ranked by $NP_{L3}$; weighted-PRE plots $P^w_{L3}(\theta_d)$, ranked by $P^w_{L3}$; and PRE plots $P_{L3}(\theta_d)$, ranked by $P_{L3}$) .

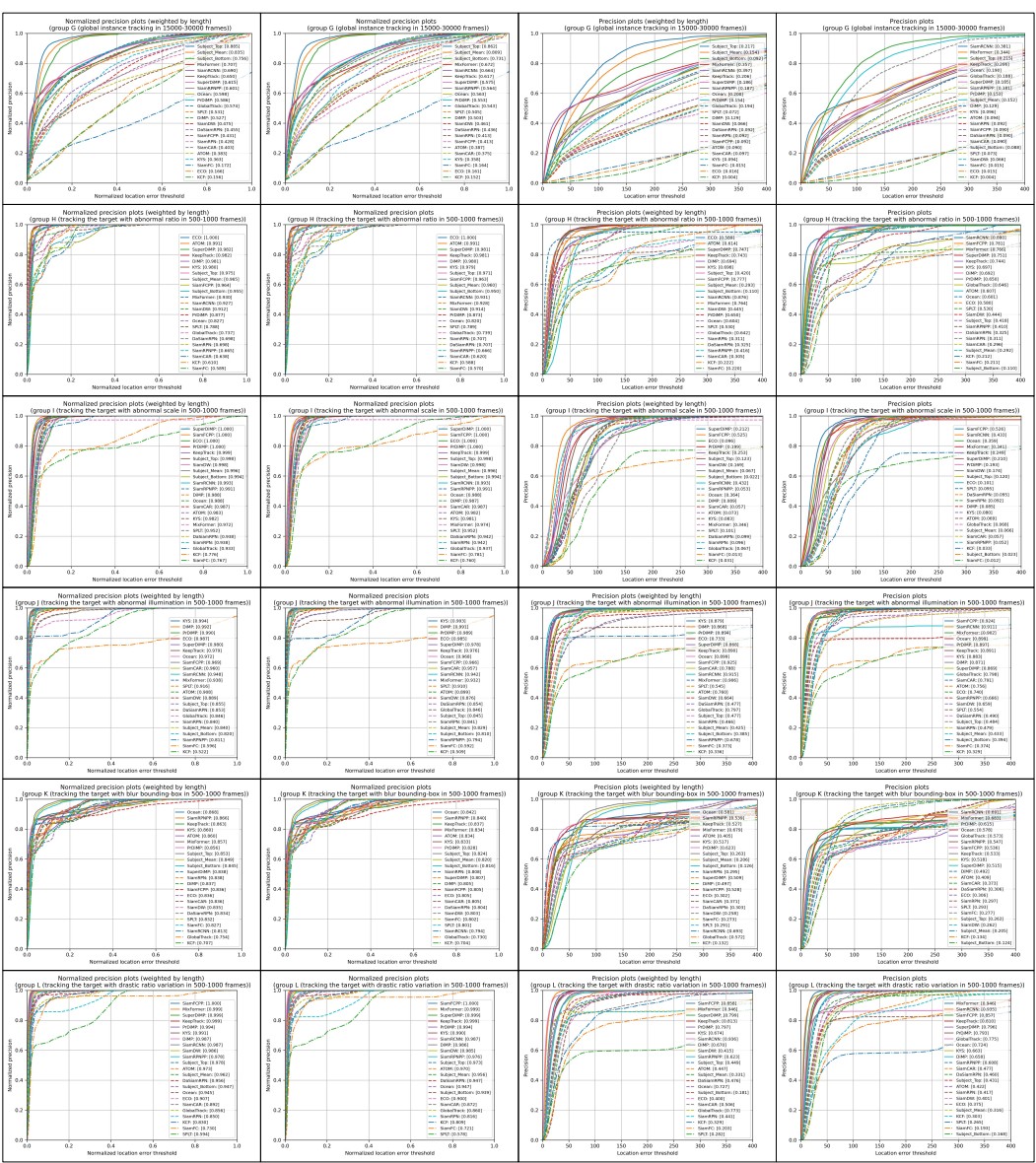

Figure 18: Results for Group G to L based on various indicators (from left to right: weighted-N-PRE plots $NP_{L3}^w(\theta_d{}')$, ranked by $NP_{L3}^w$; N-PRE plots $NP_{L3}(\theta_d{}')$, ranked by $NP_{L3}$; weighted-PRE plots $P_{L3}^w(\theta_d)$, ranked by $P_{L3}^w$; and PRE plots $P_{L3}(\theta_d)$, ranked by $P_{L3}$).

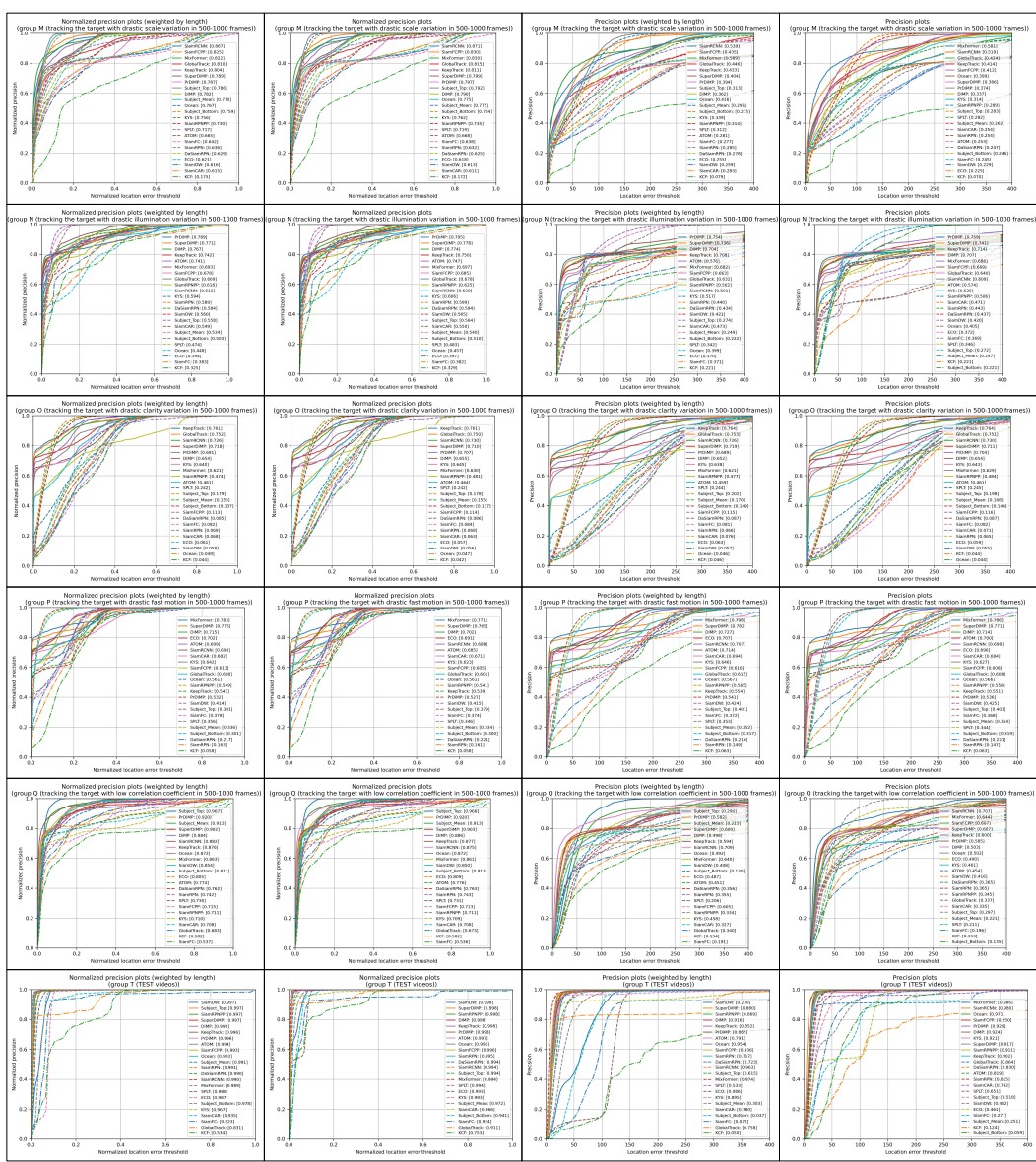

Figure 19: Results for Group M to Q and TEST videos based on various indicators (from left to right: weighted-N-PRE plots $NP^w_{L3}(\theta_d{}')$, ranked by $NP^w_{L3}$; N-PRE plots $NP_{L3}(\theta_d{}')$, ranked by $NP_{L3}$; weighted-PRE plots $P^w_{L3}(\theta_d)$, ranked by $P^w_{L3}$; and PRE plots $P_{L3}(\theta_d)$, ranked by $P_{L3}$).

## D.5 HEATMAPS

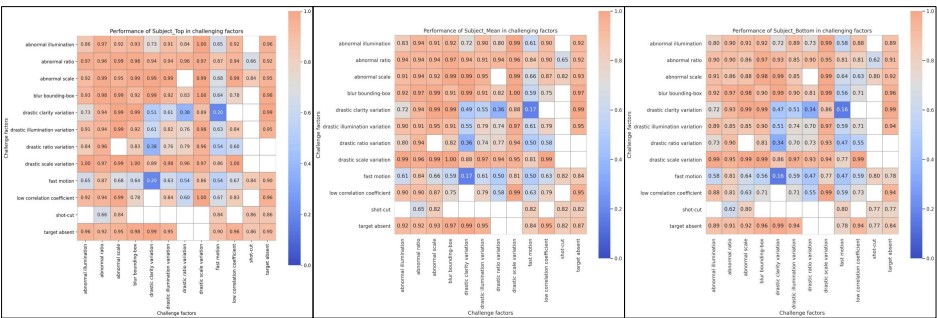

Figure 20: Performance of subjects (the maximum, mean, and minimum scores of human subjects) on multiple challenging factors based on $NP_{L2}$ .

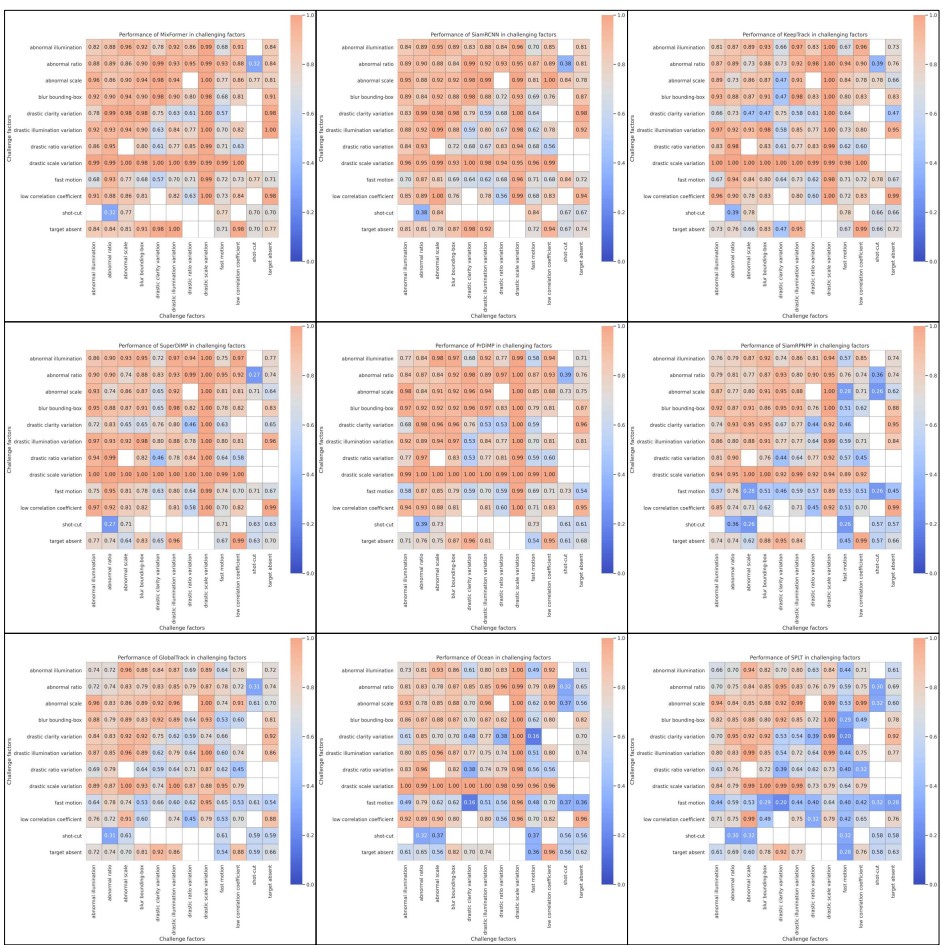

Figure 21: Performance of machines (MixFormer (Cui et al. (2022)), SiamRCNN (Voigtlaender et al. (2020)),KeepTrack (Mayer et al. (2021)), SuperDiMP (Danelljan et al. (2020)), PrDiMP (Danelljan et al. (2020)),SiamRPN++ (Li et al. (2018a)),GlobalTrack (Huang et al. (2019)), Ocean (Zhang & Peng (2020)),SPLT (Yan et al. (2019))) on multiple challenging factors based on NP$_{L2}$ .

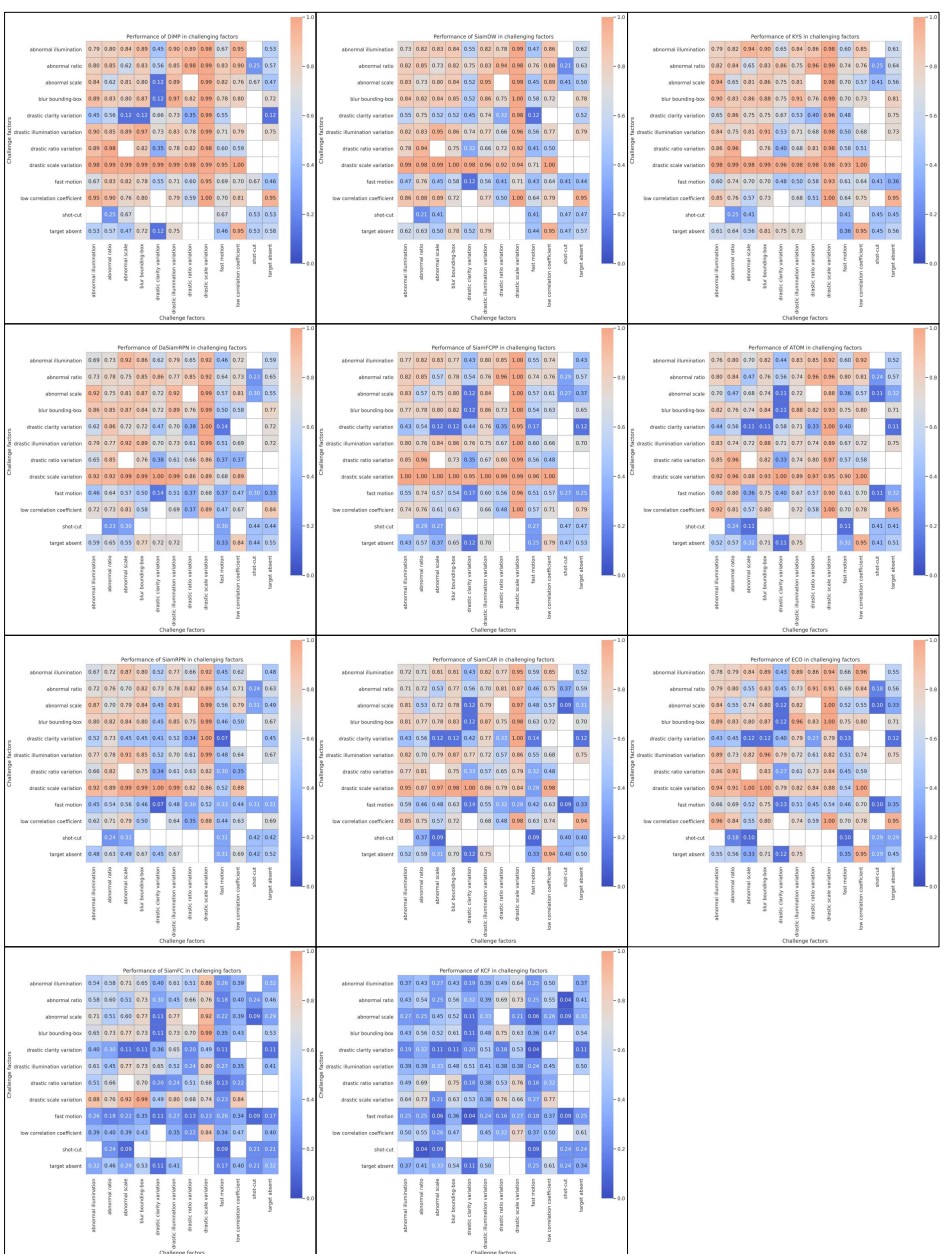

Figure 22: Performance of machines (DiMP (Bhat et al. (2019)), SiamDW (Zhang & Peng (2019)), KYS (Bhat et al. (2020)),DaSiamRPN (Zhu et al. (2018)),SiamFC++ (Xu et al. (2020)),ATOM (Danelljan et al. (2018)),SiamRPN (Li et al. (2018b)),SiamCAR (Guo et al. (2020)), ECO (Danelljan et al. (2017)),SiamFC (Bertinetto et al. (2016)),KCF (Henriques et al. (2015))) on multiple challenging factors based on $NP_{L2}$ .

## D.6 ANALYSES OF HUMAN DYNAMIC VISUAL ABILITY

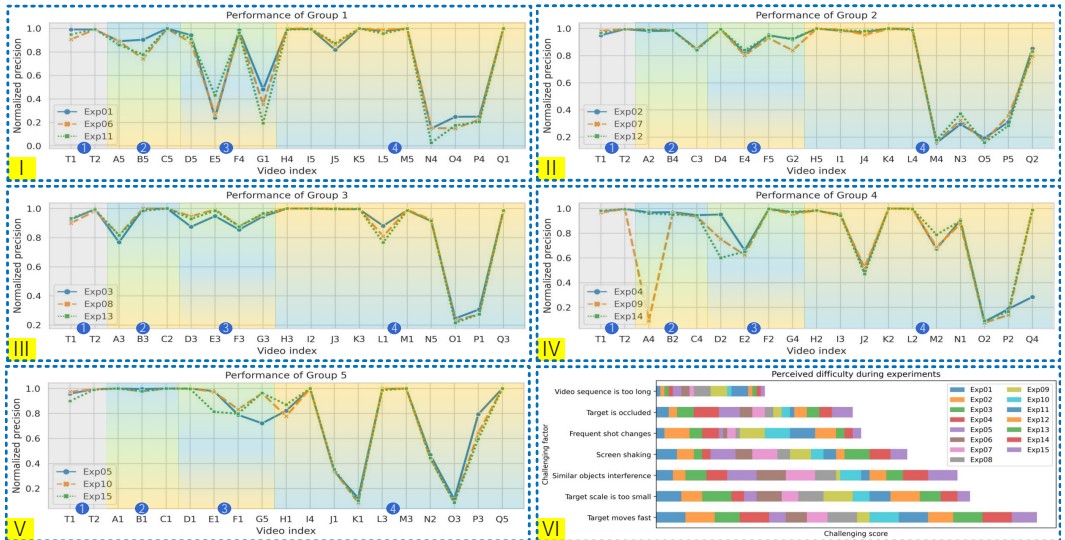

Figure 23: Scores of subject groups and self-perceptions. Subjects with a serial number interval of 5 watches the same videos in different orders. (I-V) demonstrate the performance of subjects (the vertical coordinate indicates $NP_{L2}$ score per sequence). Gray area 1 represents two TEST videos, and colorful areas 2 to 4 represent FORMAL videos. The viewing order of the subjects with blue serial number is 2-3-4, orange is 4-2-3, and green is 3-2-4. (VI) shows their self-perceptions of challenging factors in the questionnaire (a higher score means greater challenge).

We request three subjects to track the same videos in different viewing orders. Figure 23 (I-V) demonstrates that subjects' performance is independent of viewing order – watching long sequences first or last has no significant difference. However, we notice that subjects may also make mistakes, like the performance of Exp09 on A4 is significantly lower than others. A4 is a short video about a basketball game, while Exp09 misadmits the target player and keeps tracking a distractor. It indicates that humans are not absolutely infallible – they may make low-level mistakes due to carelessness.

Figure 23 (VI) shows the self-perceptions of challenging factors in the questionnaire. Most subjects consider that fast-moving targets and small targets are challenging to track, but tracking in a long sequence is not difficult – this is consistent with previous experimental conclusions in this paper.

# E PERFORMANCE OF HUMAN-MACHINE COOPERATIONS

## E.1 BOXPLOTS

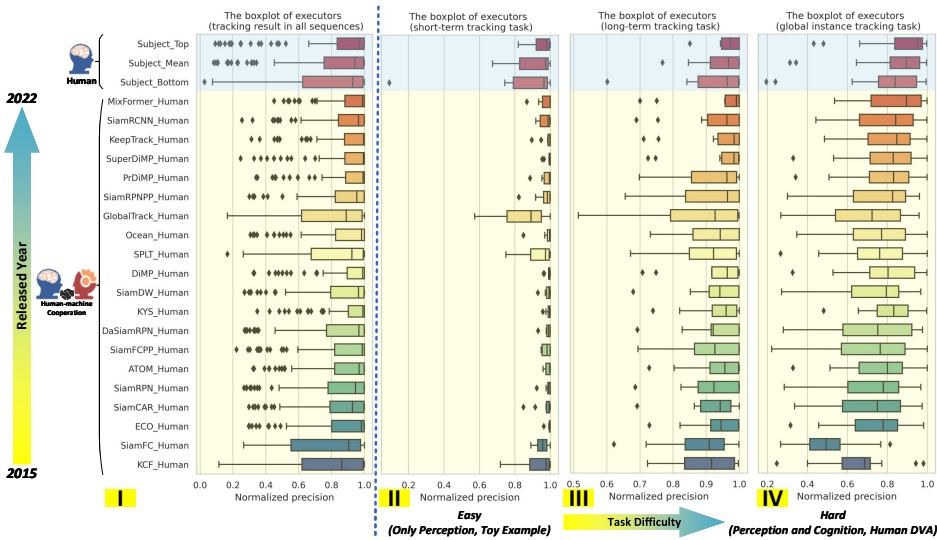

Figure 24: A comprehensive comparison of human-machine collaborators. Boxplots enumerate the distribution of $NP_{L2}$ scores for all tasks (I), short-term tracking task (II), long-term tracking task (III), and global instance tracking task (IV). The blue area represents the maximum, mean, and minimum scores of human subjects, and the yellow area represents human-machine collaborators.

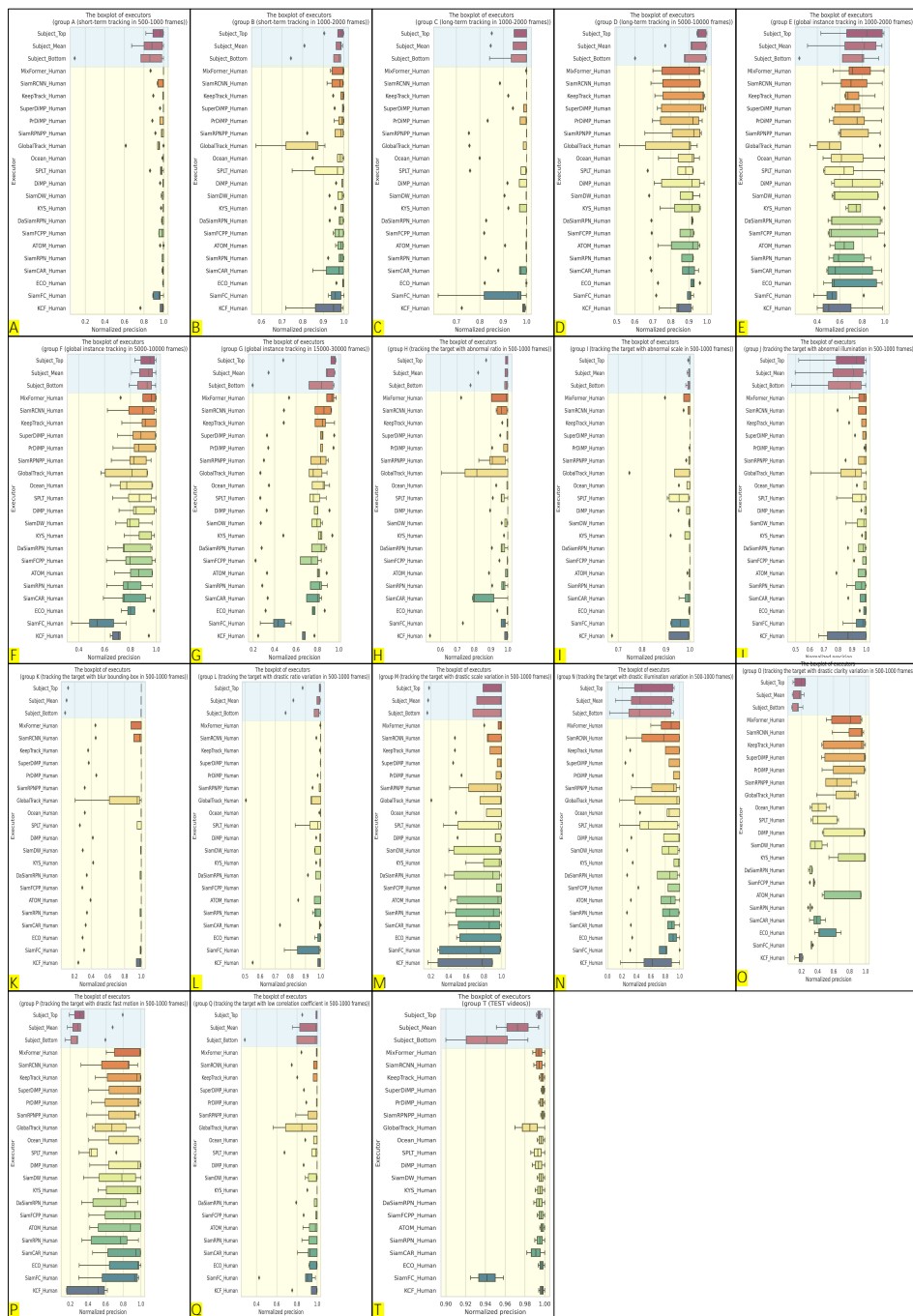

Figure 25: Boxplots enumerate the distribution of $NP_{L2}$ scores for TEST videos and Group A to Q. The blue area represents the maximum, mean, and minimum scores of human subjects, and the yellow area represents human-machine cooperation.

## E.2 PRECISION PLOTS

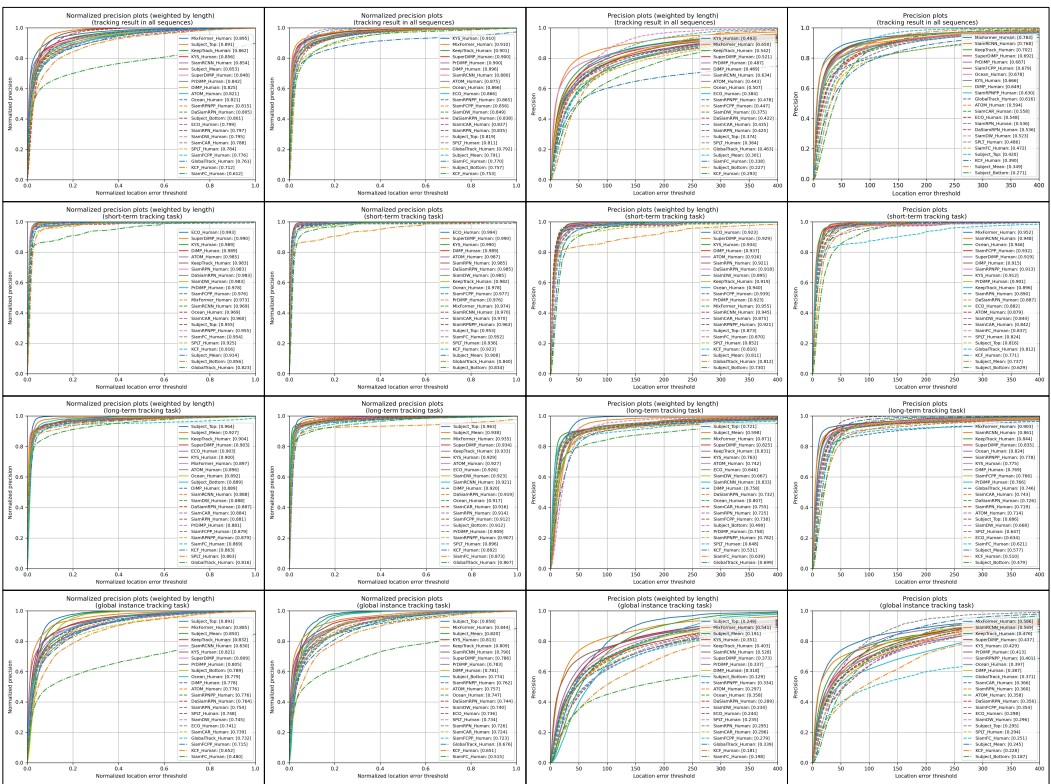

Figure 26: Results for all tasks, short-term tracking task, long-term tracking task, and global instance tracking task based on various indicators (from left to right: weighted-N-PRE plots $NP^w_{L3}(\theta_d{'})$, ranked by $NP^w_{L3}$; N-PRE plots $NP_{L3}(\theta_d{'})$, ranked by $NP_{L3}$; weighted-PRE plots $P^w_{L3}(\theta_d)$, ranked by $P^w_{L3}$; and PRE plots $P_{L3}(\theta_d)$, ranked by $P_{L3}$) .

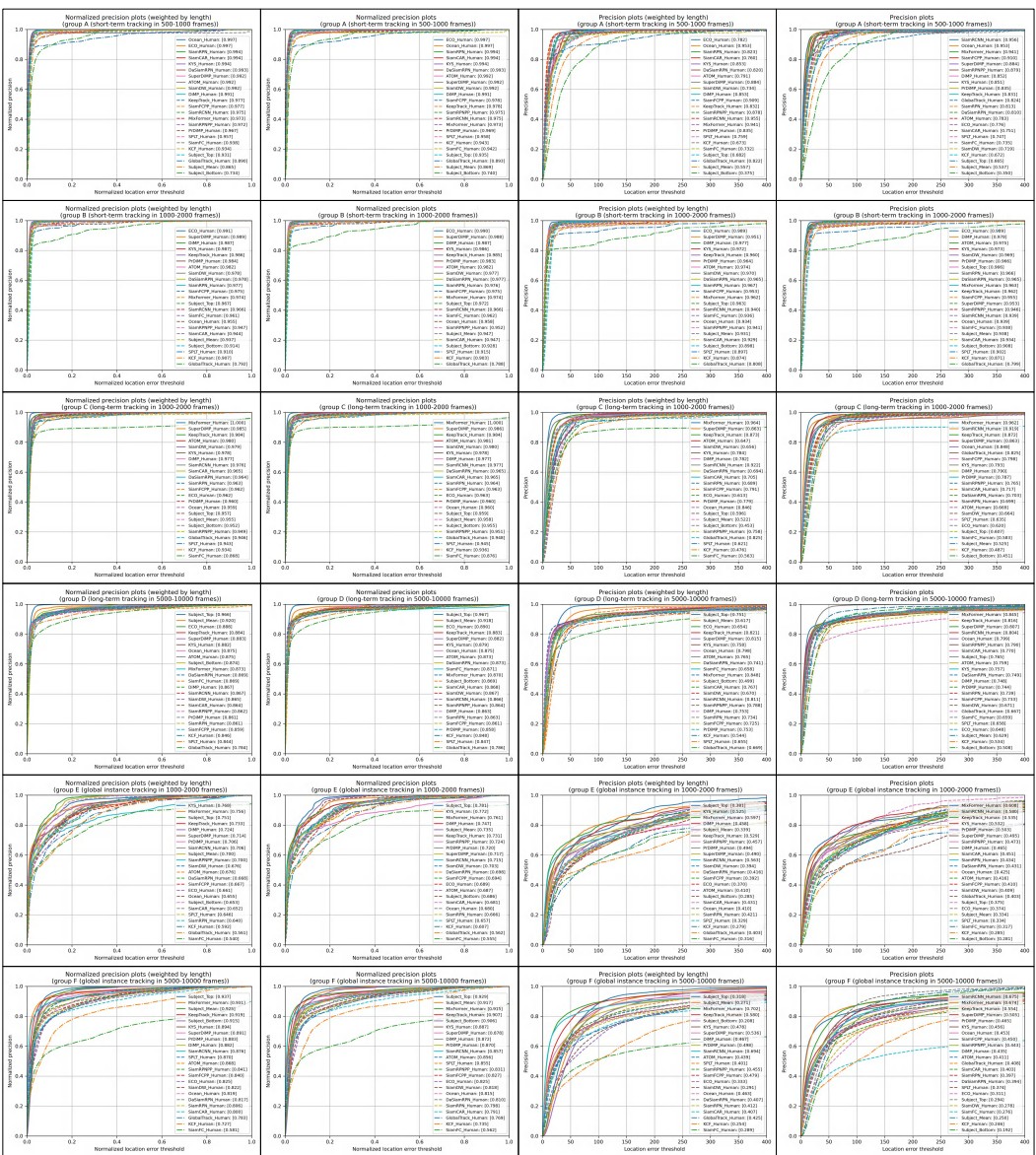

Figure 27: Results for Group A to F based on various indicators (from left to right: weighted-N-PRE plots $\text{NP}_{\text{L3}}^{\text{w}}(\theta_{\text{d}}{'})$, ranked by $\text{NP}_{\text{L3}}^{\text{w}}$; N-PRE plots $\text{NP}_{\text{L3}}(\theta_{\text{d}}{'})$, ranked by $\text{NP}_{\text{L3}}$; weighted-PRE plots $\text{P}_{\text{L3}}^{\text{w}}(\theta_{\text{d}})$, ranked by $\text{P}_{\text{L3}}^{\text{w}}$; and PRE plots $\text{P}_{\text{L3}}(\theta_{\text{d}})$, ranked by $\text{P}_{\text{L3}}$) .

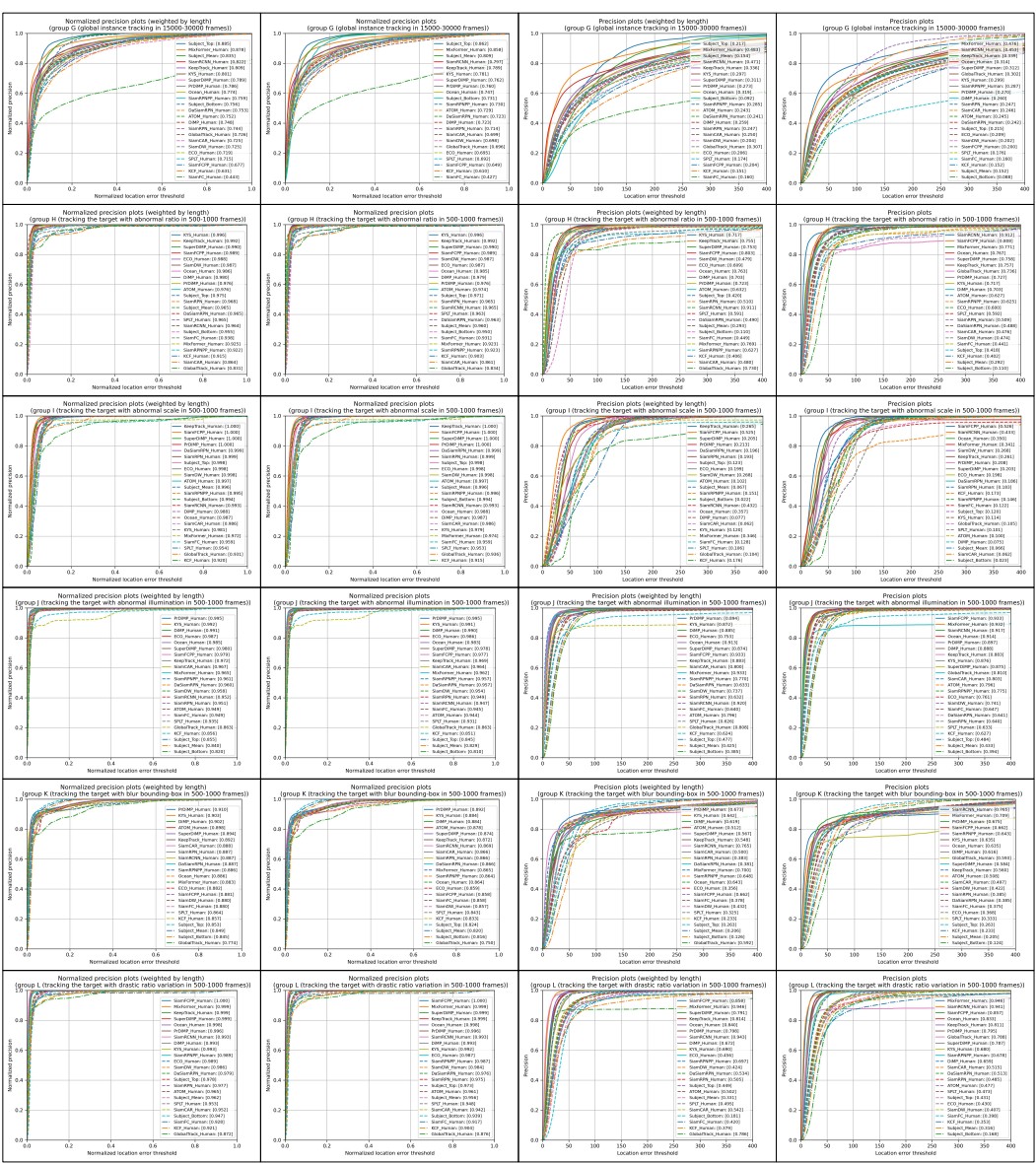

Figure 28: Results for Group G to L based on various indicators (from left to right: weighted-N-PRE plots $\text{NP}^w_{L3}(\theta_d{'})$, ranked by $\text{NP}^w_{L3}$; N-PRE plots $\text{NP}_{L3}(\theta_d{'})$, ranked by $\text{NP}_{L3}$; weighted-PRE plots $\text{P}^w_{L3}(\theta_d)$, ranked by $\text{P}^w_{L3}$; and PRE plots $\text{P}_{L3}(\theta_d)$, ranked by $\text{P}_{L3}$) .

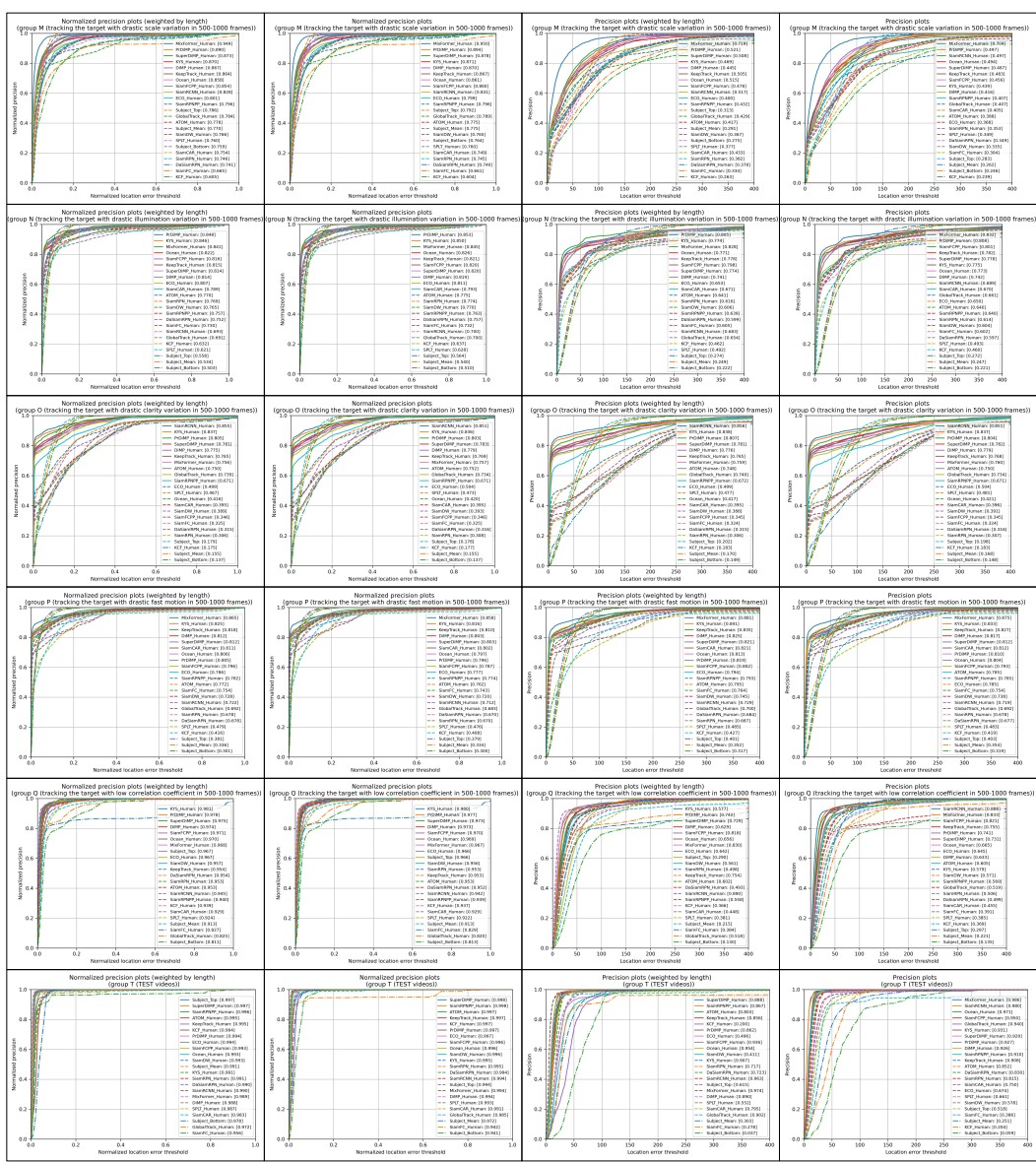

Figure 29: Results for Group M to Q and TEST videos based on various indicators (from left to right: weighted-N-PRE plots $NP^w_{L3}(\theta_d{}')$, ranked by $NP^w_{L3}$; N-PRE plots $NP_{L3}(\theta_d{}')$, ranked by $NP_{L3}$; weighted-PRE plots $P^w_{L3}(\theta_d)$, ranked by $P^w_{L3}$; and PRE plots $P_{L3}(\theta_d)$, ranked by $P_{L3}$).

## E.3 HEATMAPS

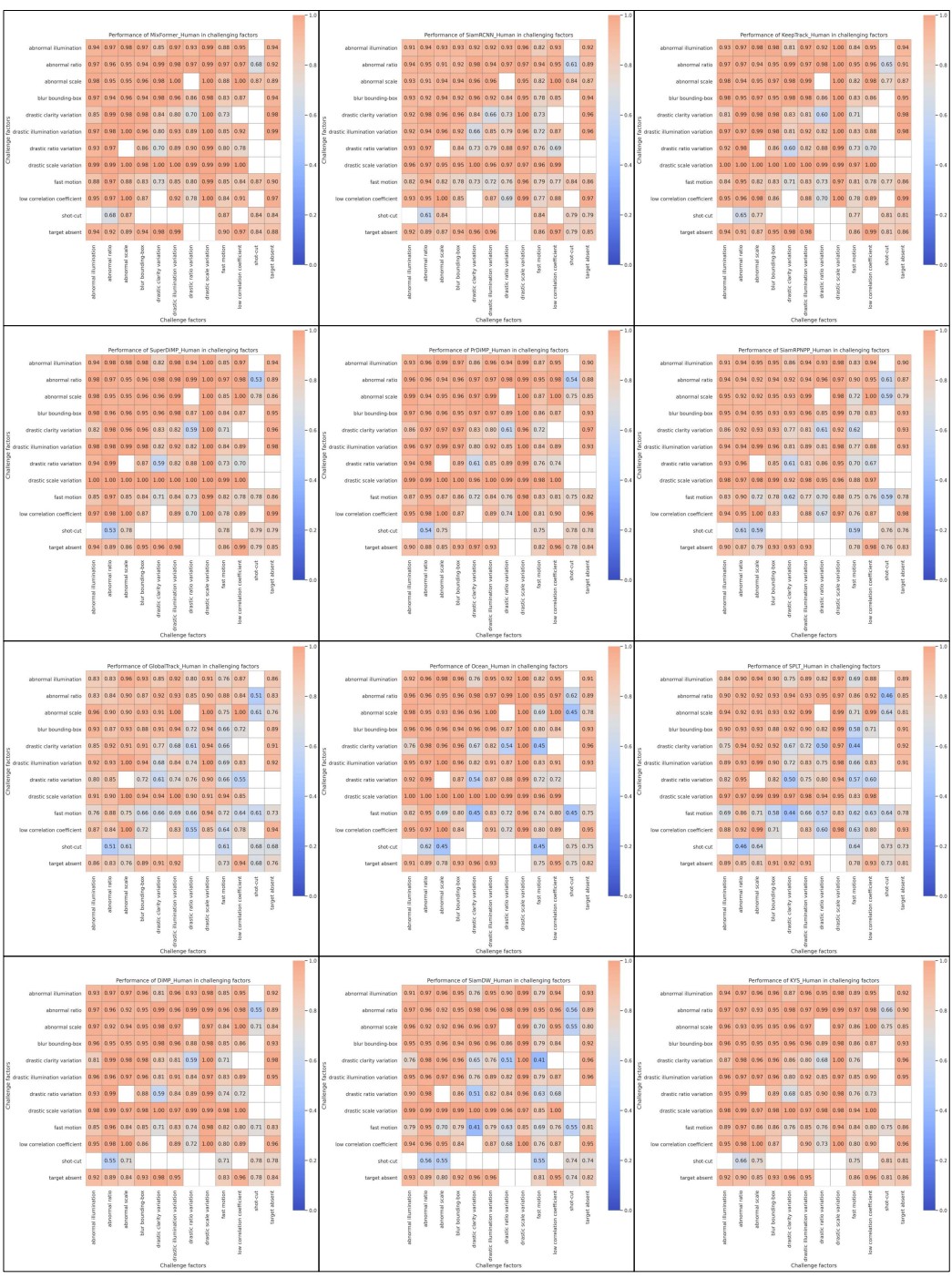

Figure 30: Performance of human-machine cooperations (MixFormer (Cui et al. (2022)), SiamRCNN (Voigtlaender et al. (2020)),KeepTrack (Mayer et al. (2021)), SuperDiMP (Danelljan et al. (2020)), PrDiMP (Danelljan et al. (2020)),SiamRPN++ (Li et al. (2018a)),GlobalTrack (Huang et al. (2019)), Ocean (Zhang & Peng (2020)),SPLT (Yan et al. (2019)), DiMP (Bhat et al. (2019)), SiamDW (Zhang & Peng (2019)), KYS (Bhat et al. (2020))) on multiple challenging factors based on $NP_{L2}$ .

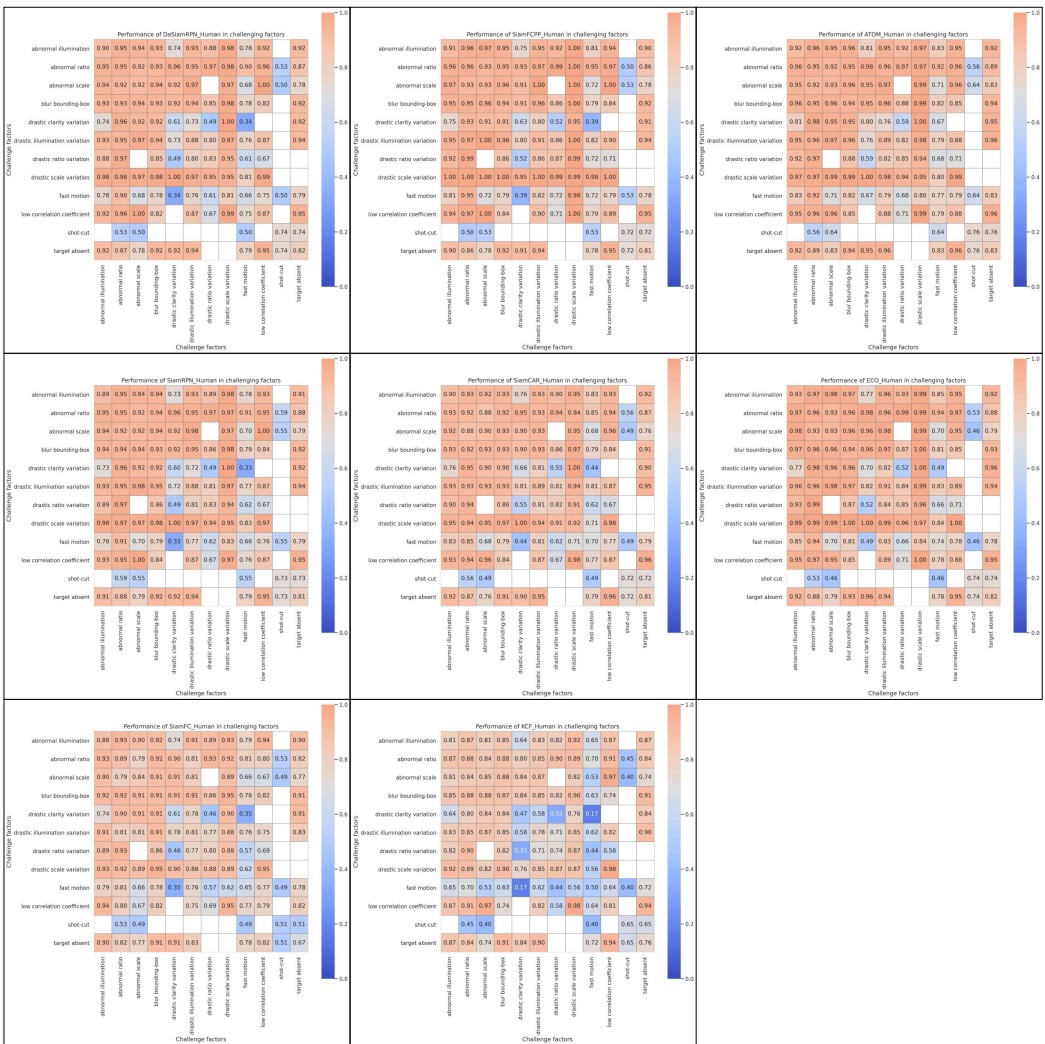

Figure 31: Performance of human-machine cooperations (DaSiamRPN (Zhu et al. (2018)),SiamFC++ (Xu et al. (2020)),ATOM (Danelljan et al. (2018)),SiamRPN (Li et al. (2018b)),SiamCAR (Guo et al. (2020)), , ECO (Danelljan et al. (2017)), SiamFC (Bertinetto et al. (2016)),KCF (Henriques et al. (2015))) on multiple challenging factors based on $NP_{L2}$ .

