# OpenReview forum: "Nearing or Surpassing: Overall Evaluation of Human-Machine Dynamic Vision Ability"
_ICLR.cc/2023/Conference — Submitted to ICLR 2023_

### Official Review · Reviewer_RdAz · 2022-10-22

**Confidence:** 3
**Correctness:** 3
**Technical Novelty And Significance:** 3
**Empirical Novelty And Significance:** 3
**Recommendation:** 3

**Clarity, Quality, Novelty And Reproducibility:**

The paper’s presentation is clear. It is of moderately good quality and originality.

**Strength And Weaknesses:**

Strength:

The paper works on an interesting problem. The presentation of this paper is clear.

The authors have provided detailed supplementary materials to support the main paper.


Weakness:

The authors used SOT as a single representative task for the evaluation system, but the concept of dynamic vision ability is much broader and includes numerous other tasks. Perhaps the authors can tune down a bit their claims throughout the paper.

Correspondingly, it is unclear to me how the results could contribute to the real applications, since the evaluation is only on SOT.

As we all know, human subjects have their biases due to their diverse cognitive background. I wonder if the authors should recruit more diversified human subjects, instead of 15 subjects while all of them are computer vision researchers aged 20-30.  Furthermore, how the machine models are trained also matters a lot. The authors can discuss the above potential issues in order to give a more convincing evaluation.


**Summary Of The Paper:**

In this paper, the authors proposed an overall evaluation system to investigate the gap between humans and machines in dynamic vision tasks. In particular, they chose single object tracking as the representative task, and design an overall evaluation system from three aspects (Experimental environment construction, Experimental executor selection, and Evaluation method design). The results provide insights for fellow researchers in the related fields.

**Summary Of The Review:**

The paper is interesting and has its merits. However, as mentioned above in the weakness part, I feel the title is a bit overclaimed and the actual contribution is not very clear.

---

> ### Author Response · Authors · 2022-11-15
> **Thanks to Reviewer RdAz and Our Reply**
>
> We especially appreciate your positive comments and valuable suggestions, which provided great courage for us to start such exploratory works. We hope the following can help clarify some of the concerns:
>
> **Q1. The authors used SOT as a single representative task for the evaluation system, but the concept of dynamic vision ability is much broader and includes numerous other tasks. Perhaps the authors can tune down a bit their claims throughout the paper. Correspondingly, it is unclear to me how the results could contribute to the real applications, since the evaluation is only on SOT.**
>
> **A1.** Thanks a lot for your concern. We have to clarify that our core contribution is primarily solving the technical difficulties and providing an evaluation framework for interdisciplinary research in the DVA area, while the SOT task is just a suitable example to demonstrate the process of this work.
> SOT is selected as a representative task because it is closer to human DVA than other vision tasks (Appendix A). And researchers can refer to our work to conduct interdisciplinary studies and evaluate human-machine DVA in more dynamic vision tasks.
> Specifically, we have listed the technical difficulties in the comprehensive interdisciplinary experiments and re-summarized our point-to-point contributions in the **Common Concern** (https://openreview.net/forum?id=LGbzYw_pnsc&noteId=bdlF0xpIrUx). Before our work, the technical difficulties described above led to fragmented research (neuroscientists aim to measure human abilities, while computer vision researchers focused only on algorithms). Our work is the first attempt to combine human and machine dynamic vision research and point out the bottlenecks of current research through detailed performance analyses of 20 representative methods, providing directions for the subsequent algorithm development.
> Besides, we have referred to your suggestions, illustrated the limitations of the work, and provided some thoughts for future research (Section 4).
>
> **Q2. As we all know, human subjects have their biases due to their diverse cognitive background. I wonder if the authors should recruit more diversified human subjects, instead of 15 subjects while all of them are computer vision researchers aged 20-30.**
>
> **A2.** Thank you for your suggestion. We follow the Small-N principle (i.e., allow a small number of subjects to perform a large number of experiments in a strict experimental setup) adopted in existing work to design our research. Of course, recruiting more subjects based on a rigorous selection process can make our study more reasonable. Therefore, we will follow your suggestion to expand the number of subjects and conduct a more detailed analysis in our future work.
>
> **Q3. Furthermore, how the machine models are trained also matters a lot. The authors can discuss the above potential issues in order to give a more convincing evaluation.**
>
> **A3.** Thank you for your suggestion. Ｗe have revised Table 2 to provide architectures and characteristics with their performance scores, and added introduce all machines used in the experiments with the technical details (characteristics, training sets, parameters, etc.) in Appendix B.

---

### Official Review · Reviewer_zwYU · 2022-10-24

**Confidence:** 3
**Correctness:** 3
**Technical Novelty And Significance:** 1
**Empirical Novelty And Significance:** 2
**Recommendation:** 3

**Clarity, Quality, Novelty And Reproducibility:**

**Clarity:**

* Figure 1(b). What is the table for? Where is the table referred to?

* In human-machine cooperation (Section 4.2), when machines fail to track, did the authors ask humans to provide the current target position? In other words, did the human subjects wait until the machine failed and then pick the position whenever the machine failed? Also, how many subjects participated in the cooperation experiments?

**Novelty:**

As written in the above section, it is not clear about the contributions of this paper.

**Reproducibility:**

The authors said in the introduction that they would open-source the toolbox, code, and metrics.



**Strength And Weaknesses:**

**Strengths:**

* The paper is well written, and the reviewer enjoyed reading it.

* The authors provide details about the experimental settings so that the readers can understand the experiments well.

* The authors conducted deep analysis for comparing human and machine DVA.

**Weaknesses:**

* The paper lacks contributions. The authors collect additional human subjects' responses for the SOT task and run 20 models to get machine responses, which is followed by deep analysis to compare the human and machine DVA. However, this reviewer is unclear what are the contributions (especially technical contributions) of this paper.

* Section 2.2 does not contain the related work on the computer vision tasks, but the section rather looks like it introduces the term. Indeed, the tasks were already presented in Section 1.

* In Figure 11: "the blue sequences" looked like they indicated the letters with a blue background, i.e. all the sequences. It is recommended either change the text background color or make it transparent. Further, it would be better to call the sequences A, B, C, ... instead of B2, D2, G4, ...  for better readability unless B2, D2, G4, ... has meanings.

* Typos:

"Aritciture" in Table 1

**Summary Of The Paper:**

With the observation that no other work compared humans and machines in dynamic vision ability, the authors designed an overall evaluation system followed by previous work. Specifically, they evaluated humans and machines on the single object tracking task and compared their performance in multiple aspects. The analysis found that humans have stronger DVA, but the gap is not big, especially in the SOTA models. Finally, they show when human-machine cooperate, the performance can be even better than humans.

**Summary Of The Review:**

The paper is well written with deep analysis to compare humans and machines in dynamic vision ability. However, the paper lacks novel contributions on the topics of learning representation.

---

> ### Author Response · Authors · 2022-11-15
> **Thanks to Reviewer zwYU and Our Reply**
>
> We especially appreciate your positive comments and valuable suggestions, which provided great courage for us to start such exploratory works. We hope the following can help clarify some of the concerns:
>
> **Q1. This reviewer is unclear what are the contributions (especially technical contributions) of this paper.**
>
> **A1.** Thanks a lot for your concern. We have listed the technical difficulties in the comprehensive interdisciplinary experiments and re-summarized our point-to-point contributions in the **Common Concern** (https://openreview.net/forum?id=LGbzYw_pnsc&noteId=bdlF0xpIrUx). Before our work, the technical difficulties led to fragmented research (neuroscientists aim to measure human abilities, while computer vision researchers focused only on algorithms). Our work is the first attempt to combine human and machine dynamic vision research and point out the bottlenecks of current research through detailed performance analyses of 20 representative methods, which provides a direction for the subsequent algorithm development.
>
> **Q2. Section 2.2 does not contain the related work on the computer vision tasks, but the section looks like it introduces the term. Indeed, the tasks were already presented in Section 1.**
>
> **A2.** Thanks a lot for your concern. We have followed your suggestion and adjusted the relevant content from the related work into the introduction and method section, which helps the readers to understand the background information of the work better and avoids repetition.
>
> **Q3. In Figure 11: "the blue sequences" looked like they indicated the letters with a blue background. It is recommended to either change the text background color or make it transparent; and typos:"Aritciture"**
>
> **A3.** Thanks a lot for your concern. We have checked grammar and spelling, and corrected the description in the figure caption based on your kind suggestion.
>
> **Q4. Figure 1(b). What is the table for? Where is the table referred to?**
>
> **A4.** We have changed Figure 1(b) to Table 1 in the revision and referred to it in the introduction (Section 1) and environment description (Section 2.1). This table is detailed information on environment settings, including task characteristics, corresponding abilities, and group information.
>
> **Q5. In human-machine cooperation (Section 4.2), when machines fail to track, did the authors ask humans to provide the current target position? Also, how many subjects participated in the cooperation experiments?**
>
> **A5.** Thanks a lot for your concern. More detailed information about our cooperation mechanism is illustrated in Appendix C.7. Here, we briefly list the key information to answer your questions.
> It is worth noting that our human-machine cooperation experiments are a preliminary exploration, and we hope to find that cooperation is possible for all algorithms. However, for a combination of 20 algorithms and 87 videos (1740 sets), it is time-consuming to arrange human subjects to participate in 1740 cooperation experiments. Therefore, we used the following strategy to organize the experiments.
> - **(Step 1)** We first use the ground-truth as target information provided to the algorithm at cooperation frames (please refer Line 8 in Algorithm 1). Since the ground-truth of sequences are provided by professional human annotators, which can be regarded as a representative of the highest level of human in static frames. Then we record all the cooperation frames as set $I_g$ ($g$ means this set is generated by ground-truth information).
> - **(Step 2)** We show the frames in $I_g$ to human subjects, asking them to find the target in the current frame and annotate it with a bounding box, then we generate the human annotations as set $I_h$ (the length of $I_g$ is equal to $I_h$).
> - **(Step 3)** We evaluate set $I_g$ and set $I_h$ based on the proposed $NP_{L2}$ score.
> - - **(Situation 1)** If the score is higher than 0.95, we consider the human subject to have the same performance as the original data annotator. Thus, we no longer organize the cooperation experiment for this algorithm-sequence combination.
> - - **(Situation 2)** On the contrary, if the score is below 0.95, we consider that the human subject has a gap with the ground-truth. We will arrange for the human subject and the algorithm to track the sequence together from the beginning, ask the human subject to annotate the target position by bounding-box in cooperation frames, and then record the collaborator's performance (Line 8 in Algorithm 1).
>
> We find that most sequences satisfied Situation 1. As the example in Figure 12, the human subject watches the first frame and clearly understands that he should locate the blue-clothed player. Then he can identify the target at frames #1125 and #6353, even if he does not watch the middle frames. This phenomenon also demonstrates the solid cognitive and memory abilities of humans. Besides, the human subjects with the best performance in a sequence will cooperate with algorithms.

---

### Official Review · Reviewer_co2u · 2022-11-03

**Confidence:** 5
**Correctness:** 4
**Technical Novelty And Significance:** 2
**Empirical Novelty And Significance:** 3
**Recommendation:** 6

**Clarity, Quality, Novelty And Reproducibility:**

The work is well structured, clear, and supported by extensive experiments. The code will be made publicly available, so it will be possible to reproduce results.

**Strength And Weaknesses:**

The claims of the paper are supported with experiments performed for multiple models. The topic is introduced well and compared with relevant studies in the same area. The DVA capabilities are examined from static and dynamic perspective. The work flows logically.

Overall the work is interesting, the only concern is comparison of centers of objects. It is not clear how you ensure humans correctly mark centers of objects. It might be difficult for humans to visually find the object center. Have you encountered any issues there? If yes, then comparison of machine's performance might not be fair. Another minor issue is complexity of figures. It's difficult to quickly understand them, and careful analysis is needed. It would be nice to simplify them if possible. It might be also beneficial to compare the proposed human-machine cooperation mechanism with other existing human in the loop approaches.

**Summary Of The Paper:**

The presented work focuses on evaluation of the performance of machines and humans in perceptual and cognitive components of dynamic visual ability and understanding the gap between humans and machines capabilities. In addition, an approach for machine and human collaboration is proposed and tested.
The experiments were performed on a wide set of models and input data showing that perceptual ability of both humans and machines are closer than the cognitive ability, in which humans outperform machines. It has been also proved that human machine cooperation is possible and can lead to improved performance.

**Summary Of The Review:**

The work is interesting and claims are supported by extensive experiments. The addressed problem and approach seems novel, even though it's using existing models/metrics.

---

> ### Author Response · Authors · 2022-11-15
> **Thanks to Reviewer co2u and Our Reply**
>
> We especially appreciate your positive comments and meaningful suggestions, which provided great courage to start such exploratory works. We summarize your questions and list answers here:
>
> **Q1. The only concern is comparison of centers of objects. It is not clear how you ensure humans correctly mark centers of objects. It might be difficult for humans to visually find the object center.**
>
> **A1.** Thanks a lot for your concern. In fact, we have considered this issue. We regard the predicted center points falling into the ground-truth box as success tracking, and calculate the proportion of success frames in the sequence as the executor's score during the evaluation ($NP_{L2}$ score, please refer to Section 2.3 and Figure 2 for detailed calculation). Therefore, for NP series indicators ($NP_{L2}$, $NP_{L3}$，$NP_{L3}^w$), the subject's prediction result falls into the ground-truth box is enough and does not have to keep the same position as the target center. Of course, we also provide the traditional metrics based on the absolute distance for analyses.
>
> **Q2. Another minor issue is complexity of figures. It's difficult to quickly understand them, and careful analysis is needed. It would be nice to simplify them if possible.**
>
> **A2.** Thanks a lot for your concern. We have simplified the figures, only preserving the most critical and visual results for the main paper, and added a description of indicators in the figures' caption.
>
> **Q3. It might be also beneficial to compare the proposed human-machine cooperation mechanism with other existing human in the loop approaches.**
>
> **A3.** Thanks a lot for your concern. We provide a detailed description of the cooperation mechanisms in Appendix C.7, explaining that our experiments only explore the possibility of cooperation through a simple and intuitive mechanism, and there is still much space for further optimization. Given that the human-machine collaboration in single object tracking is still inadequate, we will refer to your suggestions in our subsequent research, compare our work with existing human-in-the-loop approaches in other research areas, and further refine the collaboration mechanism.

---

### Author Response · Authors · 2022-11-15
**Rebuttal Summary**

We especially appreciate all reviewers for their valuable comments and suggestions. All of the concerns have been addressed with the following updates. The adaptations in our revision are as follows:

(1)	**ABSTRACT:** We optimize the presentation of methods contributions to make it easier to understand.

(2)	**INTRODUCTION (Section 1):** We add Figure 1 to compare the proposed evaluation framework with existing works; list the technical difficulties in the human-machine experiments and re-summarize our point-to-point contributions.

(3)	**RELATED WORK:** We follow the suggestion of Reviewer zwYU and adjust the relevant content from the related work into the introduction and method section, which helps the readers to understand the background information of the work better and avoids repetition.

(4)	**ENVIRONMENT (Section 2.1):** We enumerate DVA works in neuroscience and add the differences of our work in the experimental environment setting.

(5)	**EXECUTOR (Section 2.2):** For 20 machines used in this work, we revise Table 2 to provide architectures and characteristics with their performance scores.

(6)	**EVALUATION (Section 2.3):** To accurately evaluate the performance of executors, we divide the evaluation dimensions into three granularities (frame-level, sequence-level, and group-level), add the equations, and draw Figure 2 to show the detailed information for each granularity.

(7)	**EXPERIMENTS (Section 3):** We refer to the Reviewer co2u 's suggestion to simplify the figures, only preserving the most critical and visual results for the main paper, and add a description of indicators in the figures' caption.

(8)	**CONCLUSIONS AND FUTURE WORK (Section 4):** We explain why we use the SOT task as a proxy for DVA, illustrate the limitations of the work, and provide some thoughts for future research.

(9)	**TASK DESCRIPTION (Appendix A):** We add the comparison of vision tasks (Appendix A.1) and the representative sequence of SOT (Section A.2).

(10)	**MACHINES (Appendix B.2):**　Ｗe introduce all machines used in the experiments and add the technical details (characteristics, training sets, parameters, etc.).

(11)	**COOPERATION MECHANISM (Appendix C.7):** We use pseudo code to illustrate the framework of the cooperation mechanism, and add the details of human-machine cooperation experiments.

Besides, we have checked grammar and spelling, and supplemented the description of indicators in all figures' captions. Please refer to the revised version for detailed information; all the updates have been highlighted in blue color.

---

### Author Response · Authors · 2022-11-15
**Common Concern**

We especially appreciate all reviewers for their valuable comments and suggestions. All reviewers believe this work is interesting and well-written with a logical structure. However, several reviewers have some concerns about the contributions.

We regret that the original version mainly focuses on describing and analyzing the experimental results and neglects the technical difficulties, resulting in the article's contributions being unclear.

In the revised version, we have listed the technical difficulties in the comprehensive interdisciplinary experiments and re-summarized our point-to-point contributions:

(1)	**Experimental environment construction.** The first difficulty in environment construction is *compatibility*. Choosing a high-contrast toy environment used in classical neuroscience work is too simple to evaluate DVA accurately. On the other hand, human psychophysical experiments are expensive and time-consuming, and cannot be assessed on large-scale datasets like machines. Based on this, the second difficulty is *representativity*. With the limitation of dataset scale, the environment should not only fully represent the characteristics and difficulties of DVA, but also provide a graded experimental setup for subsequent analyses. To entirely reflect the task characteristics and thoroughly compare the human-machine DVA, we choose 87 videos with 244,455 frames to construct the environment. All videos are carefully picked from various dimensions, ensuring the environment can cover the perceptual and cognitive components of DVA.

(2)	**Experimental executor selection and record.** As an interdisciplinary experiment, the task executors involve humans and machines. The difficulty for the human aspect is *accurately recording their performance in dynamic vision tasks* and ensuring that the recorded results can be quantified to support subsequent evaluations. Through detailed comparison and analysis, we select the mouse-based method and design a toolbox for human subjects to record their frame-level performance in real-time. For machines, since we aim to study the DVA gap between humans and machines, there are two concerns in model selection: *trends* (i.e., selecting various algorithms) and *gap* (i.e., focusing on the upper bound of algorithms). Thus, both classic and SOTA methods with different architectures are selected to explore whether the research route has effectively shortened the gap. Consequently, task executors included 20 representative algorithms and 15 subjects (all subjects are computer vision researchers aged 20-30). All experiments are managed in a strict process.

(3)	**Evaluation metrics design.** Traditional indicators usually select the positional relationship (e.g., intersection over union (IoU) and center distance) between the predicted result and ground-truth to accomplish calculation. However, the tracking result of humans is a point, which cannot be calculated by IoU. Thus, we use two center points (i.e., the target center predicted by executors, and the center point of ground-truth) to design three granularities (frame-level, sequence-level, and group-level) for evaluation. Especially, we consider the influence of sequence length and revise the group-level indicators to generate a more appropriate evaluation. Experimental analyses have verified the validity of the proposed metrics.

The 87 sequences with frame-level human-machine comparison and cooperation results, the toolbox for recording real-time human performance, codes for sustaining various evaluation metrics, and evaluation reports for 20 representative models will be open-sourced to help researchers develop intelligent research on dynamic vision tasks.

---

### Author Response · Authors · 2022-11-18
**Looking Forward to Your Feedback**

Dear all Reviewers,

Thanks for your review, which helps us spot and resolve the ambiguities. We hope that we have addressed your concerns in this Rebuttal. As it is the last day of Discussion Stage 1, if you have any other questions, please feel free to let us know.

Best regards,
Authors of #359

---

### Decision · Program_Chairs · 2023-01-20

**Decision:**

Reject

**Justification For Why Not Higher Score:**

Overall, the proposed benchmark and associated human baselines failed to convince the reviewers and the AC thus recommends the paper to be rejected.

**Justification For Why Not Lower Score:**

N/A

**Metareview: Summary, Strengths And Weaknesses:**

This paper received 2 rejects and 1 borderline accept and while during discussion one reviewer remained somewhat positive about the work the paper lacks support. The paper describes a new set of synthetic visual object tracking tasks together with human benchmarks.

In general, the 2 negative reviewers challenged the contributions of this work and the AC agrees with them.  It makes sense to build computational models of biological vision and to compare their performance against human observers but this is not what this work is about. The models are just AI models and they are not biologically plausible in any way.

Conversely, since the goal for these systems is to achieve the best possible accuracy and humans are far from perfect on these tasks why should AI researchers care about these human baselines? A clear demonstration of the benefits of this artificial challenge for real-world tracking in videos is needed.